# Violation of detailed balance in non-equilibrium magnons observed by inelastic neutron scattering

Chengyun Hua [1] ✉, Barry L. Winn [2], Colin Sarkis[2], Gabriele Sala[3], Takeshi Egami [1,4,5,6] & David A. Tennant[4,5,6] ✉

Traditional inelastic neutron scattering (INS) characterizes excitations—such as phonons and magnons—in condensed matter systems at thermodynamic equilibrium. However, the most intriguing and puzzling many-body effects in open quantum systems often emerge from dissipative dynamics that are inherently out of equilibrium. Here, we use a combination of laser pumping and INS to experimentally observe long-lived nonequilibrium magnons in a two-dimensional (2D) square-lattice Heisenberg antiferromagnet. These nonequilibrium magnons manifest themselves as a violation of detailed balance in the dynamic structure factor and reach steady states under periodic driving, analogous to nonequilibrium steady states in driven dissipative systems. Furthermore, we show that the violation of detailed balance reflects the quantum-mechanical nature of the underlying dynamical system, where out-of-time-ordered correlations of creation and annihilation operators do not satisfy commutation relations. The *in operando* INS technique developed here provides a new approach to studying nonequilibrium magnons in prototypical 2D quantum magnets and can be extended to other systems, including one-dimensional spin chains and topological many-body spin systems, where nonequilibrium effects are widespread and rich in discovery potential.

Quantum spin systems provide a rich platform for exploring the fundamental principles of nonequilibrium physics in condensed matter. The ubiquitous presence of integrability in one-dimensional (1D) spin chains—a property of dynamical systems with an infinite set of conserved commuting quantities—gives rise to intriguing nonequilibrium phenomena, such as quantum wake dynamics[1], finite Drude weight[2,3], and relaxation to nonequilibrium steady states (NESS)[4–7]. In higher-dimensional magnets—such as two-dimensional (2D) or three-dimensional (3D) antiferromagnets—the Bose-Einstein condensation of magnons has been observed under parametric microwave pumping[8–11]. These condensed states of bosonic excitations hold promise for enabling dissipationless—and potentially entangled—information

transport[12,13]. The direct detection of quantum states and their thermalization processes in nonequilibrium spin systems offers invaluable insights into the fundamental mechanisms governing nonequilibrium many-body physics.

To experimentally probe quantum states and their dynamical evolution, advanced spectroscopic techniques are essential. Among them, inelastic neutron scattering (INS) stands out as one of the most powerful tools for studying microscopic spin states. INS measures the momentum- and energy-dependent magnetic structure factor, $S(\mathbf{Q}, E)$, which reflects spin-spin correlations. The momentum- and energy-resolved measurements of spin dynamics using low-energy, non-invasive neutrons give INS distinct advantages over other techniques—

[1]Materials Science and Technology Division, Oak Ridge National Laboratory, Oak Ridge, TN, USA. [2]Neutron Scattering Division, Oak Ridge National Laboratory, Oak Ridge, TN, USA. [3]Japan Proton Accelerator Research Complex, Tokai Ibaraki, Japan. [4]Department of Materials Science and Engineering, the University of Tennessee, Knoxville, TN, USA. [5]Department of Physics and Astronomy, the University of Tennessee, Knoxville, TN, USA. [6]Shull Wollan Center, Oak Ridge National Laboratory, Tennessee, USA. ✉e-mail: huac@ornl.gov; dtennant@utk.edu

such as angle-resolved photoemission spectroscopy[14], inelastic x-ray scattering[15], optical spectroscopies[16], and device transport measurements[10]–in uncovering novel phenomena in spin systems.

Most importantly, neutron scattering from spins is a quantum mechanical process whose scattering cross section directly probes the annihilation and creation of spin excitations, *i.e.* magnons (quantized spin waves). In a magnon annihilation process, a neutron scatters off an excited state with energy $E$, returns it to the ground state, and gains that energy. Conversely, in a magnon creation process, a neutron loses energy $E$ and excites the system to a state $E$ above the ground state. In thermodynamic equilibrium, the principle of detailed balance requires that magnon annihilation processes are equilibrated by magnon creation processes[17–19]. Consequently, the dynamic structure factor measured by INS must satisfy the relation $S(\mathbf{Q}, -E) = \exp(-E/k_B T)S(\mathbf{Q}, E)$, where $k_B$ is the Boltzmann constant, and $T$ is the system temperature[20].

When the underlying scattering system is driven out of equilibrium, the principle of detailed balance is no longer satisfied in magnon creation and annihilation processes. This violation signifies a breakdown of microreversibility and can serve as a quantitative measure of the system's deviation from equilibrium[21]. The extent to which detailed balance is broken reflects the transition rates between microscopic states, providing valuable insight into the decay behavior of nonequilibrium excitations. In this work, we demonstrate that quantifying the degree of detailed balance violation in inelastic neutron scattering (INS) measurements enables time-resolved access to the dynamics of nonequilibrium magnons in $Rb_2MnF_4$, a model quasi-two-dimensional square-lattice Heisenberg antiferromagnet.

This work is enabled by a recent advance in laser-pump, neutron-probe techniques, in which a nanosecond-pulsed laser periodically excites magnons into nonequilibrium states, and inelastic neutron scattering (INS) probes their subsequent dynamics (conceptual schematic shown in Fig. 1a). This unique capability–developed at HYSPEC (Hybrid Spectrometer, beamline 14-B; Picture of the capability in

operation shown in Fig. 1b) at the Spallation Neutron Source, Oak Ridge National Laboratory–opens a new direction for time-resolved neutron spectroscopy, allowing momentum-resolved studies of nonequilibrium excitations in quantum materials. Full experimental details are provided elsewhere[22], with a brief description of the crystal and measurement setup in the Methods Section.

## Results and discussion

We report three key findings. First, we observe photo-induced nonequilibrium magnon populations in $Rb_2MnF_4$ via INS, evidenced by a clear violation of detailed balance between magnon creation and annihilation processes. Second, the system evolves into a nonequilibrium steady state (NESS) under periodic excitation, facilitated by the intrinsic conservation laws of the magnon system and by weak magnon-phonon coupling. Third, this violation reveals the quantum mechanical nature of the underlying dynamics. Using quantum transport theory, we calculate the out-of-time-order commutator between creation and annihilation operators, $a^\dagger(t)$ and $a(t)$, providing a microscopic explanation for the breakdown of detailed balance in a driven-dissipative quantum system.

To observe non-equilibrium magnon dynamics, it is essential to first establish an equilibrium baseline for the INS measurements. Figure 2a shows the measured magnetic intensity, $I(\mathbf{Q}, E)$, of $Rb_2MnF_4$ along the $H$ direction in the $[HHL]$ scattering plane at 3.6 K. Since magnon transport in $Rb_2MnF_4$ is confined to the $ab$ plane, no dispersion is observed along the $L$ direction. To generate a two-dimensional intensity map, the measured $I(\mathbf{Q}, E)$ was integrated over $L \in [2.1, 2.9]$ reciprocal lattice units (r.l.u.). For a Heisenberg-type exchange interaction, as in $Rb_2MnF_4$, the off-diagonal spin-spin correlation functions vanish, *i.e.* $S^{\alpha\beta} = 0$ for $\alpha \neq \beta$, and the longitudinal component $S^{zz}$ is much weaker than the transverse components due to large spin moments ($S = 5/2$). Therefore, the total dynamic structure factor is well approximated by $S(\mathbf{Q}, E) \approx g_x^2 S^{xx}(\mathbf{Q}, E) + g_y^2 S^{yy}(\mathbf{Q}, E)$, where $g_{x,y}$ are the relevant g-factors. Based on a Heisenberg

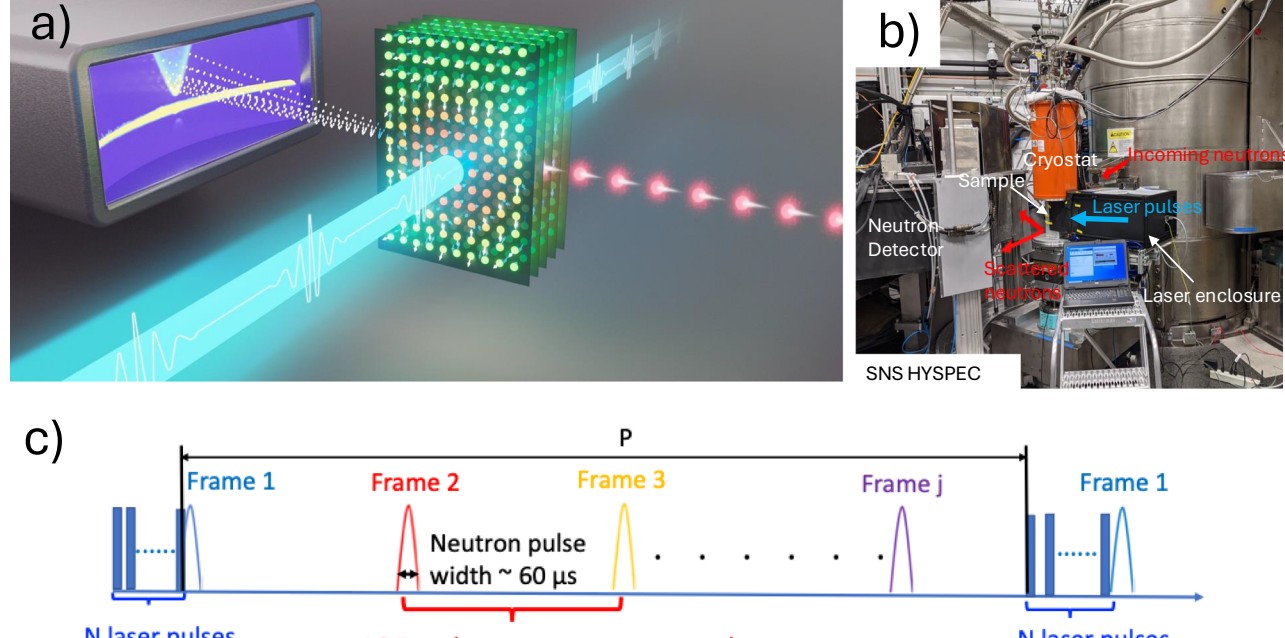

**Fig. 1 | Experimental setup. a** Conceptual schematic of a laser-pump neutron-probe technique. A pulsed laser is used to excite the spin system into non-equilibrium states, and then the resulting dynamics of spin excitations will be probed by inelastic neutron scattering. **b** Picture of the laser setup integrated into the HYSPEC beamline with a neutron-compatible optical cryostat installed.

**c** Schematics of a sequence of laser pulses ($N$: laser pulse number) triggered by an external signal that is synchronized to the neutron spallation events with a tunable repetition frequency ($j$: neutron pulse number). The width of the neutron pulse is about 60 µs.

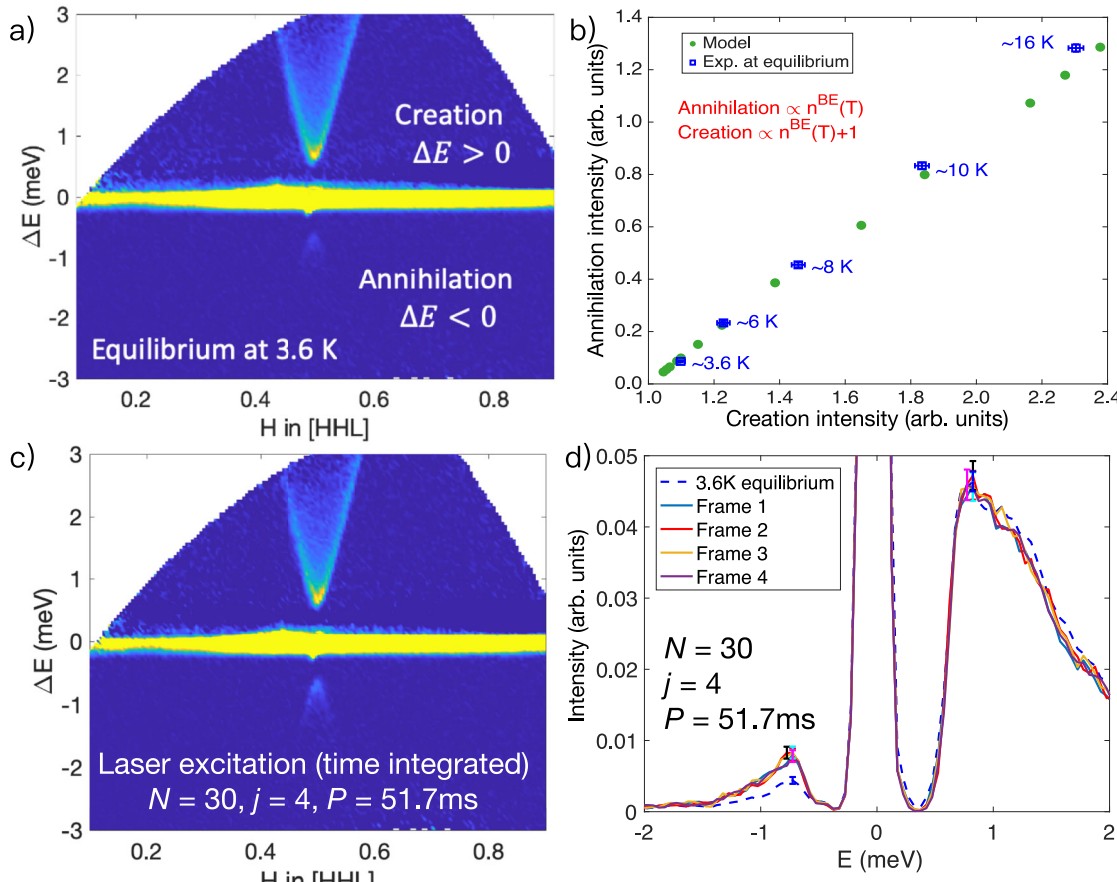

**Fig. 2 | INS measurements at equilibrium and under laser excitation. a** Measured magnetic intensity $I(\mathbf{Q}, E)$ of Rb$_2$MnF$_4$ along $H$ in [$HHL$] plane at equilibrium (3.6 K). Data are integrated over $L \in$ [2.1, 2.9] r.l.u. A magnetic anisotropy gap of 0.6 meV is observed. **b** Magnon annihilation intensity plotted against creation intensity at various equilibrium temperatures, based on measured data (blue squares) and theoretical model (green circles). Data are integrated over $H \in$ [0.475, 0.525] and $L \in$ [2.1, 2.9] r.l.u; energy ranges are $E \in$ [ − 2, − 0.3] meV for annihilation and $E \in$ [0.3, 2] meV for creation. The linear trend with slope near unity confirms detailed balance at equilibrium. **c** Laser-excited magnetic intensity $I(\mathbf{Q}, E)$ of Rb$_2$MnF$_4$ along $H$ in [$HHL$] plane at 3.6 K. The number of laser pulses per pumping

event is set to $N = 30$ and the number of neutron pulses arriving at the sample between two laser pumping events is set to $j = 4$, resulting in $P \sim 51.7$ ms. Color scales in (**a**) and (**c**) are identical. **d** Comparison of $I(Q, E)$ with laser excitation (solid lines) and at equilibrium (blue dashed line), integrated over $H \in$ [0.475, 0.525] r.l.u. near the zone center. On the magnon creation side, the intensity remains consistent with equilibrium. On the annihilation side, however, a photoinduced excess magnon population is observed. No decay in the annihilation signal is detected across the four neutron frames. Representative error bars at ± 0.6 meV denote ± 1$\sigma$ uncertainties derived from Poisson neutron counting statistics ($\sigma = \sqrt{N}$).

Hamiltonian with anisotropy along the $z$-axis, $I(\mathbf{Q}, E)$ can be accurately predicted using linear spin-wave theory combined with McViNE simulations[23] that incorporate instrumental resolution effects[22,24].

We performed equilibrium measurements at several temperatures well below the Néel temperature of Rb$_2$MnF$_4$ ($T_N = 38$ K) (see SI Sec. IA for complete temperature-dependent INS data and their comparison with theoretical predictions). These measurements confirm that detailed balance between magnon annihilation and creation intensities is strictly obeyed at equilibrium (Fig. 2b). Specifically, the total creation and annihilation intensities exhibit a linear relationship with a slope close to unity. As the temperature approaches zero (lower-left corner of Fig. 2b), the annihilation intensity vanishes while the creation intensity approaches unity. This behavior reflects the fact that neutrons can always create magnons from the ground state, even at 0 K.

To induce nonequilibrium magnon states, we use a nanosecond pulsed laser with photon energy ~2.4 eV, pulse energy up to 300 μJ, and a fixed repetition frequency of 2000 Hz. The laser intensity, duration, and excitation period are tunable parameters. As illustrated in Fig. 1c, we define the number of laser pulses per excitation event as $N$, the number of neutron pulses between successive excitation events as $j$, and the time between excitation events as $P$. Given the fixed laser

repetition rate (2000 Hz) and the SNS operating at 60 Hz, $P$ is determined by both $N$ and $j$. For example, in the measurements shown in Fig. 2c, d, we use $N = 30$ and $j = 4$, resulting in $P \sim 51.7$ ms. Figure 2c shows the time-integrated $I(\mathbf{Q}, E)$ under this laser excitation scheme. Compared to the equilibrium case (Fig. 2a), an additional intensity on the annihilation side ($E < 0$) is observed under laser excitation.

To investigate this effect in greater detail, we integrate the signal over a narrow $H$-range around the zone center. Figure 2d, which shows data separately reduced for each neutron frame between the laser pumping events, reveals that the magnon creation side is well described by the equilibrium data, while a photoinduced excess intensity appears only on the annihilation side. As shown in SI Sec. IB, this increase in annihilation intensity is statistically significant and non-thermal in origin.

This energy-resolved analysis reveals a clear breakdown of equilibrium magnon statistics. A natural question is whether detailed balance could nevertheless hold when the magnon occupation deviates from a Bose-Einstein distribution. Although these two conditions can, in principle, be mathematically decoupled, we emphasize that for the bosonic magnon system studied here they are fundamentally linked.

Assessing the breaking of detailed balance solely through the relation, $I(\mathbf{Q}, -E) \neq \exp(-E/k_B T) I(\mathbf{Q}, E)$, is insufficient in out-of-

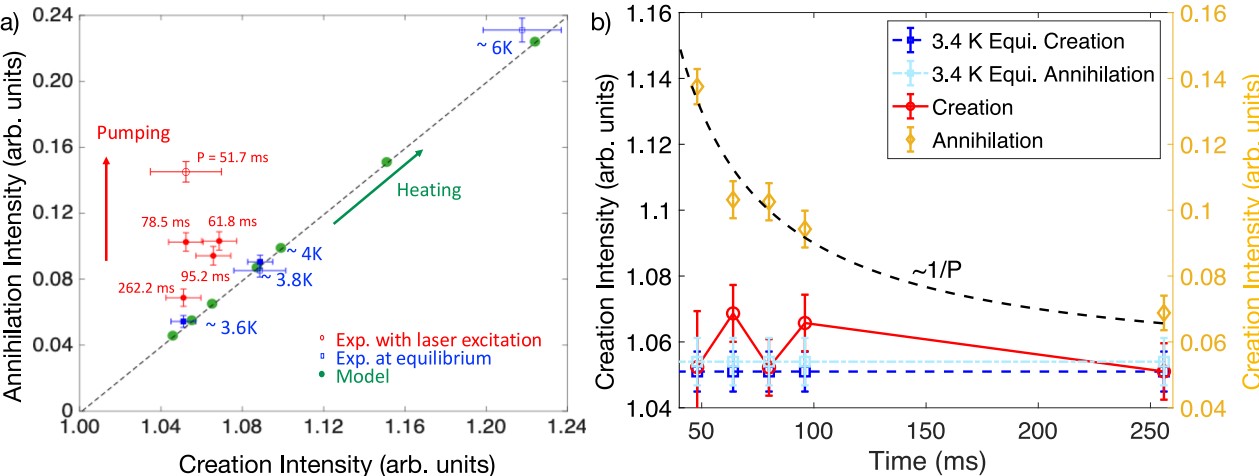

**Fig. 3 | Magnon annihilation versus creation processes under nonequilibrium conditions. a** Laser excited (open and solid red circles), equilibrium (solid blue squares), and modeled (green solid circles) magnon annihilation versus creation intensity. $H$ is integrated between [0.475, 0.525]. On the annihilation side, $E$ is integrated over the range $-2$ to $-0.3$ meV. On the creation side, $E$ is integrated from 0.3 to 2 meV. In-equilibrium data points at various temperatures fall on the linear dashed line, while pumping of non-equilibrium magnons only increases the annihilation intensity. **b** Creation (left axis; red circles) and Annihilation (right axis; yellow diamonds) intensity as a function of time obtained from five sets of INS measurements under laser excitation with different $P$. The annihilation intensity decreases inversely with $P$, indicating a nonequilibrium steady state in a driven-dissipative system. The laser excited intensities are compared to their corresponding equilibrium values at 3.6 K. All error bars denote $\pm 1\sigma$ uncertainties derived from Poisson neutron counting statistics ($\sigma = \sqrt{N}$).

equilibrium bosonic systems. When a unique thermodynamic temperature does not exist, apparent agreement with this relation can arise trivially–particularly at low temperatures, where spectral weight is concentrated near the dispersion minimum, and the measured lineshape is dominated by instrumental energy resolution rather than intrinsic magnon statistics. In this regime, fitting the intensity ratio effectively reduces to a single-point constraint, making it always possible to extract an "effective temperature" irrespective of whether the underlying magnon population is thermal. Such a temperature lacks physical meaning once the system departs from equilibrium.

In equilibrium, a bosonic magnon system is described by a Bose-Einstein distribution characterized by a single, well-defined temperature, which enforces detailed balance between magnon creation and annihilation processes. Under laser excitation, however, the magnon-creation spectra remain unchanged, while the annihilation intensity is systematically enhanced. Interpreted within an equilibrium framework, this would require mutually incompatible temperatures for creation and annihilation processes, directly demonstrating that the magnon population cannot be described by a single Bose-Einstein distribution.

Moreover, as shown in SI Sec. IB, not only the magnon-creation spectra but also the magnetic Bragg peaks remain unchanged under laser excitation, confirming that the long-range magnetic order is preserved and that the system remains fundamentally bosonic. The excess intensity on the annihilation side, therefore, reflects a non-thermal overpopulation of magnons. Since transition rates are proportional to occupation numbers, this population asymmetry implies that magnon creation and annihilation processes are no longer microscopically reversible. We thus conclude that the definition of a thermodynamic temperature becomes ill-defined under laser excitation and, consequently, that the dynamic magnetic structure factor violates the condition of detailed balance, *i.e.* $I(\mathbf{Q}, -E) \neq \exp(-E/k_B T) I(\mathbf{Q}, E)$.

Another striking feature in Fig. 2d is the absence of decay in the nonequilibrium magnon population on the annihilation side over a 51.7 ms timescale, indicating that a steady state is reached. The total time-integrated annihilation and creation intensities are compared in Fig. 3a (open red circle). While the equilibrium data follow a linear relationship consistent with detailed balance, under laser excitation, the

annihilation intensity increases significantly, whereas the creation intensity remains unchanged from the 3.6 K equilibrium value. To probe the relaxation behavior, we vary the time between laser pumping events ($P$) while keeping the energy input per pumping event constant, and monitor the intensities as a function of $P$ (solid red circles in Fig. 3a). We find no decay between pumping events, and the creation intensity remains consistent with the equilibrium distribution at 3.6 K (see SI Sec. IB for more details). As shown in Fig. 3b, the annihilation intensity decreases inversely with $P$.

The observed inverse $P$ dependence of the annihilation intensity indicates that the photoinduced magnon population evolves into a nonequilibrium steady state (NESS) under periodic excitation, in agreement with our earlier theoretical predictions[25]. Although the initial magnon states are not directly accessible, laser-pump INS measurements reveal that the excess magnon population accumulates at the lowest-energy part of the magnon spectrum—on the annihilation side of $I(\mathbf{Q}, E)$. This accumulation is enabled by the constraints of magnon-magnon scattering in a Heisenberg antiferromagnet, which must satisfy multiple conservation laws.

Due to magnetic anisotropy, the leading higher-order term in the expansion of a Heisenberg Hamiltonian is symmetric in spin operators, *i.e.* $aa^\dagger aa^\dagger$. This symmetry restricts intrinsic magnon-magnon interactions in $Rb_2MnF_4$ to elastic pairwise collisions of magnons, conserving crystal momentum, energy, and particle number[26]. Through these interactions, within a few microseconds at 3.6K[24], high energy nonequilibrium magnons rapidly decay to the lowest energy states, establishing a non-equilibrium magnon distribution of form, $[\exp((E - \mu(t))/k_B T) - 1]^{-1}$, where $\mu(t)$ is a time-dependent chemical potential. As outlined in SI Sec. II, this steady-state distribution arises from Boltzmann transport theory.

The decay of $\mu(t)$ is determined by non-conserving interactions—primarily magnon-phonon scattering—which occur on a much longer timescale than magnon-magnon interactions. In gapped systems such as $Rb_2MnF_4$ at low temperature, magnon-phonon scattering is expected to be weak due to limited phase space overlap between magnon and phonon modes. The timescale inferred from our measurements suggests that magnon-phonon relaxation occurs on the order of hundreds of milliseconds. Such long lifetimes of nonequilibrium magnons are consistent with prior observations of magnon

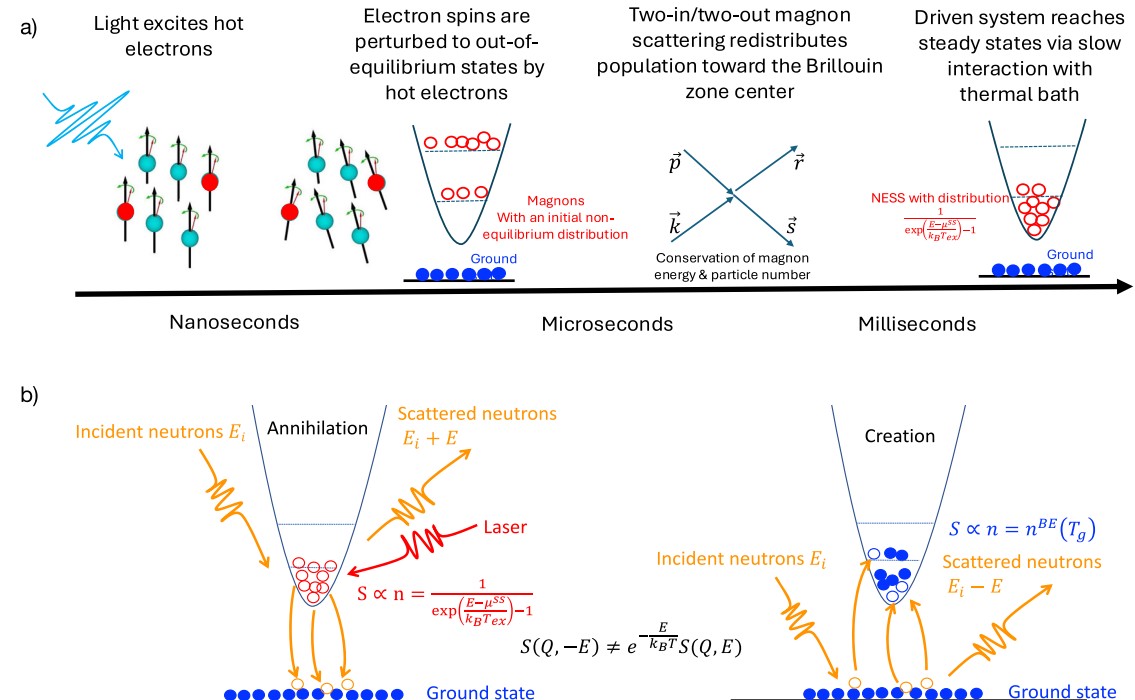

**Fig. 4 | Illustration of out-of-equilibrium events during INS. a** Timeline of physical events in a laser-neutron pump-probe experiment. The intense pulsed laser induces hot electrons, and their spins are then affected, leading to non-equilibrium spin states within nanoseconds. Through elastic pairwise collisions of magnons, within a few microseconds at low temperatures, high energy non-equilibrium magnons decay to the lowest energy states with a non-equilibrium magnon distribution in the form of $[\exp((E - \mu^{ss})/k_B T_{ex}) - 1]^{-1}$, where $\mu^{ss}$ is a chemical

potential. **b** Illustration of neutron annihilation and creation processes under laser pumping conditions. The population distribution of the excited states results from laser excitation, while the ground state remains intact and remains in equilibrium with the thermal bath. When neutrons annihilate magnons, they detect the population distribution of non-equilibrium magnon states induced by the laser. When neutrons create magnons from the ground state, they create a canonical ensemble average of magnons according to the temperature of the ground state.

Bose-Einstein condensation in CsMnF$_3$[27], a Heisenberg antiferromagnet with similar characteristics to Rb$_2$MnF$_4$.

This separation of relaxation timescales–illustrated in Fig. 4a–facilitates the observed non-equilibrium steady state. While electron and phonon subsystems thermalize on submicrosecond to microsecond scales (verified by an undistorted magnon-creation spectrum and thermal modeling), the magnon subsystem remains decoupled due to significantly slower relaxation. In gapped Heisenberg antiferromagnets like Rb$_2$MnF$_4$, number-conserving two-in/two-out magnon scattering redistributes population toward the Brillouin zone center but cannot restore a thermal distribution. The spectral gap acts as a relaxation bottleneck, deferring equilibration to the much longer timescales of magnon-phonon coupling. Because the driving period is shorter than this coupling time, the system enters a steady state governed by conservation laws rather than lattice temperature. Consequently, the violation of detailed balance is an intrinsic feature of the driven magnon population, independent of initial excitation inhomogeneities.

One of the most intriguing features of the out-of-equilibrium measurements is that the nonequilibrium magnon population manifests exclusively as a change in intensity on the annihilation side of $I(\mathbf{Q}, E)$. The unbalanced changes between magnon creation and annihilation processes suggest that the canonical ensemble average of the spin correlation function may be altered under nonequilibrium conditions. For instance, the ground and excited states may each be described by distinct canonical ensemble distributions. A simplified physical picture is illustrated in Fig. 4b: the excited-state population is shaped by laser excitation, while the ground state remains in equilibrium with the thermal bath.

In this framework, magnons are annihilated from a nonequilibrium steady-state distribution, $[\exp((E - \mu^{ss})/(k_B T_{ex})) - 1]^{-1}$ while they are created into an equilibrium distribution, $[\exp(E/(k_B T_g)) - 1]^{-1}$. But why do we require two distinct ensemble averages to describe these processes? While the violation of detailed balance under nonequilibrium conditions is well established[28], the microscopic mechanism responsible for this imbalance remains unclear. What breaks the reciprocity between creation and annihilation processes, and how does the system partition itself into coexisting thermal and driven subsystems?

Boltzmann theory, as a classical transport framework, captures the NESS distribution of excited magnons under experimental conditions, it offers only part of the picture. Specifically, it fails to explain the deeper quantum mechanism underlying the emergence of two distinct canonical ensemble averages for magnon creation and annihilation processes. The inherently quantum mechanical nature of these processes calls for a quantum transport theory to fully describe their dynamical behavior under nonequilibrium conditions.

To establish such a quantum framework, the magnetic structure factor, $S(\mathbf{Q}, E)$, is first expressed within linear spin-wave theory in terms of creation and annihilation operators, $a^\dagger$ and $a$. In thermal equilibrium, the time-varying form of the operators $a_l(t)$ and $a_l^\dagger(t)$ are given as a Fourier expansion of normal modes:

$$a_l(t) = N^{-1/2} \sum_{\mathbf{Q}} \exp\{i(\mathbf{Q} \cdot \mathbf{l} - \omega_{\mathbf{Q}} t)\} a_{\mathbf{Q}}, \tag{1}$$

$$a_l^\dagger(t) = N^{-1/2} \sum_{\mathbf{Q}} \exp\{-i(\mathbf{Q} \cdot \mathbf{l} - \omega_{\mathbf{Q}} t)\} a_{\mathbf{Q}}^\dagger, \tag{2}$$

where $\langle a_{\mathbf{Q}}^\dagger a_{\mathbf{Q}} \rangle = n_{\mathbf{Q}}^{BE}(T)$ gives the annihilation intensity and $\langle a_{\mathbf{Q}} a_{\mathbf{Q}}^\dagger \rangle = n_{\mathbf{Q}}^{BE}(T) + 1$ gives the creation intensity. When the underlying Hamiltonian is time-dependent and the dynamical system is a driven-dissipative open system—such as the NESS observed in Rb$_2$MnF$_4$—the time evolution of $a_l(t)$ and $a_l^\dagger(t)$ is no longer governed by Eqs. (1) & (2).

In such cases, a quantum Langevin equation (QLE) must be employed to describe the temporal evolution of physical observables under the combined influence of external driving and coupling between different modes. However, modeling quantum transport in mixed states of an interacting many-body system remains an open and formidable challenge. A complete treatment of this problem lies beyond the scope of the present work. Instead, we demonstrate the violation of detailed balance within a simplified toy model governed by the following Hamiltonian:

$$\mathcal{H} = \mathcal{H}_0 + \hbar\omega_b b^\dagger b + \hbar\omega_s \delta a^\dagger \delta a + \hbar\lambda(\delta a^\dagger + \delta a)(b^\dagger + b) \quad (3)$$

where $\mathcal{H}_0 = \hbar\omega_s a_0^\dagger a_0$ is the equilibrium Hamiltonian such that $\langle a_0^\dagger a_0 \rangle = n^{BE}(T)$. $\delta a$ is the first order expansion of $a$ around $a_0$ such that the structure factor can be written as

$$S(\omega) = S_0(\omega) + \delta S(\omega) \propto \langle a_0^\dagger a_0 \rangle \delta(\omega + \omega_s) + \langle a_0 a_0^\dagger \rangle \delta(\omega - \omega_s)$$
$$+ \frac{1}{2\pi} \int_{-\infty}^{\infty} (\langle \delta a(-\omega') \delta a^\dagger(-\omega) \rangle + \langle \delta a^\dagger(\omega') \delta a(\omega) \rangle) d\omega'. \quad (4)$$

Equation (3) describes two coupled single-mode harmonic oscillators: a "spin" mode, $\hbar\omega_s \delta a^\dagger \delta a$, and a "bath" mode, $\hbar\omega_b b^\dagger b$, representing spin and lattice degrees of freedom, respectively. The parameter $\lambda$ denotes the coupling strength between the two modes; this coupling hybridizes the original degrees of freedom and gives rise to new collective quasiparticles that characterize the long-range interacting system. Only after identifying these eigenmodes do we introduce external pumping of the quasiparticles and their damping at a rate $\gamma$, which governs slow relaxation toward thermal equilibrium. Although highly simplified, this minimal toy model captures the essential physics of driven dissipative dynamics at a heuristic level.

Diagonalizing the Hamiltonian yields a new low-energy normal mode with eigenfrequency, $\tilde{\omega}_s(\omega_s, \omega_b, \lambda)$, with corresponding annihilation and creation operators $c$ and $c^\dagger$ (See SI Sec. III). We identify this $c$-mode with the low-energy mode observed in our INS measurements. This simplified treatment is justified in Rb$_2$MnF$_4$ by the clear separation of timescales: magnon-phonon coupling occurs much more slowly than the dominant spin and bath interactions, allowing the normal mode approximation to capture the relevant physics.

In the toy Hamiltonian, we neglect fast interactions—such as magnon-magnon scattering—and focus solely on the damping of the $c$-modes due to coupling with a thermal bath, characterized by a damping rate $\gamma$. The mathematical derivation is very similar to the calculation of the dynamic structure factor for inelastic photon scattering from a dilute quantum gas undergoing a nonequilibrium structural phase transition[29].

To model the generation of $c$-modes by fast microscopic processes induced by external pumping, we introduce an input operator $c_{in}(t)$ in QLE, with correlation functions $\langle c_{in}^\dagger(t) c_{in}(t') \rangle = 0$ and $\langle c_{in}(t) c_{in}^\dagger(t') \rangle = \delta(t - t')$. By solving the QLE, Solving the QLE under these conditions reveals that the out-of-time-ordered correlation between $\delta a(t)$ and $\delta a^\dagger(t)$ does not commute, indicating a fundamental breakdown of micro-reversibility in the driven-dissipative system. Under the assumption $\omega_s/\omega_b < 1$, the non-equilibrium correction to the structure factor is analytically obtained as:

$$\delta S(\omega) \propto \frac{1}{2\pi} \left( \frac{\tilde{\omega}_s}{\omega_s} + \frac{\omega_s}{\tilde{\omega}_s} \right) \frac{\gamma}{(\omega + \tilde{\omega}_s)^2 + \gamma^2}. \quad (5)$$

with a width determined by the damping rate $\gamma$. This peak corresponds to the generation of $c$-modes and manifests as excess intensity on the annihilation side of the dynamic structure factor. The physical picture behind Eq. (5) is remarkably intuitive: what is created is eventually annihilated. In this sense, the dynamical system remains "balanced" in terms of creation and annihilation processes.

More sophisticated quantum modeling and simulation will be required to fully understand many-body quantum effects out of equilibrium in laser-neutron pump-probe experiments. For now, as a proof-of-principle demonstration, the observed violation of detailed balance between magnon creation and annihilation processes under laser pumping in a model quantum magnet represents a crucial first step. It establishes that neutron scattering can probe microscopic aspects of nonequilibrium dissipative dynamics in open quantum systems.

It will be particularly interesting to explore whether similar violations of detailed balance can be observed in other two- or three-dimensional Heisenberg magnets that satisfy the conditions for forming a nonequilibrium steady state as discussed above. More broadly, the development of *in operando* inelastic neutron scattering platforms opens a previously unexplored frontier: momentum- and energy-resolved studies of dissipative quantum dynamics at milli-electron-volt energy and microsecond time scales[30]. These capabilities are especially powerful in one-dimensional and frustrated spin systems, where thermal equilibration of out-of-equilibrium spin states is predicted to be extremely slow—or, in some cases, never completed. This includes open questions such as fractal dynamics of magnetic monopoles in spin ice, anomalous transport in 1D spin chains, and nonequilibrium enhancement of many-body entanglement in cuprates. Advancing our understanding of these phenomena will have broad implications, ranging from novel memory devices to coherent transport in quantum materials for low-loss, low-power electronics, and controllable entanglement in quantum matter. This work sets the stage for addressing fundamental challenges in nonequilibrium quantum physics using neutron-based methods.

## Methods

Measurements were performed using the HYSPEC direct geometry spectrometer (beamline 14-B) at the Spallation Neutron Source (SNS). The experiment used a single crystal of Rb$_2$MnF$_4$, a spin-5/2 quasi-two-dimensional square-lattice Heisenberg antiferromagnet. The orthorhombic crystal ($a=b=4.28$ Å, $c=14$ Å) had dimensions of approximately $10 \times 8$ mm$^2$ in the ab-plane and a thickness of 3 mm along the c-axis. The sample was aligned with the [$HHL$] plane as the horizontal scattering plane and was housed in a modified optical cryostat (CRYO-B) equipped with 2-inch outer quartz windows and 70-mm inner sapphire windows, maintaining a base temperature of 3.4 K via continuous liquid helium flow. The incident neutron energy ($E_i$) was set to 5 meV. Due to interference with the laser enclosure and beamline components, the sample rotation angle ($s_1$) was restricted to $-45° < s_1 < +25°$, limiting $L \in [1, 4]$.

Out-of-equilibrium magnons were generated using a Coherent Flare NX pulsed laser operating at 515 nm ($\sim 2.4$ eV) with a 5 ns pulse width and a maximum pulse energy of 300 μJ. The laser power was strictly controlled using a motorized half-wave plate and a polarizing beam splitter to prevent sample overheating. A transient grating optical setup was employed: the laser passed through a phase mask and two 4-inch lenses (focal length ratio 1:2) to cross two time-coincident coherent pulses at the sample. This formed a spatial interference pattern with a 2 μm periodicity and an enlarged beam diameter of 5 mm at the sample. The optical axis remained orthogonal to the sample's ab-plane throughout the measurements. The laser was

externally triggered and synchronized with the 60 Hz neutron spallation pulses using TTL signals.

## Data availability

The data that support the findings of this article are openly available[31].

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

## Acknowledgements

The authors would like to acknowledge Andrei T. Savici for time-split neutron data processing, Vladislav Sedov for laser-neutron synchronization firmware development, Chris Redmon for CRYO-B modifications, David Connor for mechanical mounting and alignment, and Mariano Ruiz-Rodriguez for setting up the graphical user interface that manages ADCROC timing settings. Support for inelastic neutron scattering, McViNE simulation, data analysis, and modeling was supported by the U.S. Department of Energy, Office of Science, Basic Energy Sciences, Materials Sciences and Engineering Division. Magnon theory was supported by the U.S. Department of Energy, Office of Science, National Quantum Information Science Research Centers, Quantum Science Center. A portion of this research used the resources at the Spallation Neutron Source, supported by DOE, BES, Scientific User Facilities Division.The beamtime was allocated to HYSPEC on proposal numbers IPTS-27068 and IPTS-30516. This manuscript has been authored by UT-Battelle, LLC, under contract DE-AC05-00OR22725 with the US Department of Energy (DOE). The US government retains, and the publisher, by accepting the article for publication, acknowledges that the US government retains a non-exclusive, paid-up, irrevocable, worldwide license to publish or reproduce the published form of this manuscript, or allow others to do so, for US government purposes. DOE will provide public access to these results of federally sponsored research in accordance with the DOE Public Access Plan (http://energy.gov/downloads/doe-public-access-plan).

## Author contributions

C.H. and D.A.T. conceived the project and co-wrote the manuscript with input from all other authors. C.H., B.L.W., and C.S performed the inelastic neutron scattering experiments. G.S. performed the McViNE simulation. C.H. and T.E. analyzed the neutron data. D.A.T developed the theoretical model.

## Competing interests

The authors declare no competing interests.
