## [Peer Review File · Nature Communications]

Violation of detailed balance in non-equilibrium magnons observed by inelastic neutron scattering

Corresponding Author: Dr Chengyun HUA

Version 0:

Reviewer comments:

Reviewer #2

(Remarks to the Author)

How materials respond when driven out of equilibrium and then recover is intimately tied in with their functionality. This manuscript reports the production of a non-equilibrium steady-state of the collective magnetic excitations – that is, spin waves – induced by periodic laser pumping and probed with inelastic neutron scattering. The experimental novelty is a pump-probe methodology implemented by the authors, reported in their Ref. 22, to create and study the non-equilibrium state. Furthermore, the manuscript develops a comprehensive theoretical analysis to explain the origin of the non-equilibrium state and the observed experimental data in terms of the Boltzmann transport equation.

Inelastic neutron scattering (INS) is the premier technique to study the wavevector and frequency dependence of excited states in condensed matter; its application to the study of non-equilibrium states is in its very infancy. The work reported here is original and will be of significance and interest to researchers working in magnetism and in non-equilibrium states. The only work that comes to mind which combines driving a magnetic material into a non-equilibrium state which is then probed with INS is Reeder et al PNAS 122 e2415300121 (2025), which use microwaves tuned to a specific transition in a molecular magnet – a very different pumping method than here, which has very different physics internal to the probed material that results in a non-equilibrium but steady state.

The present work is impressive and complete in its novel (with neutrons) methodology, and the theoretical interpretation laid out in the Supplementary Information (SI). However, having been very positive to this point, there are significant questions I have about an important part of the analysis of the experimental data as they are presented, namely convincingly demonstrating that the magnon spectra when the sample is laser-pumped data cannot be described by detailed balance. This is key to the whole assertion that the sample is in a non-equilibrium steady state, and therefore needs to be carefully addressed. Unfortunately, what appear to be editing errors in the Supplementary Information – where actually a great deal of essential information is placed – mean that the argument is incompletely presented, and which serve to raise a series of questions. The authors must convincingly address these points before I can consider recommending acceptance. I have a number of lesser points which are detailed below. Otherwise, the work is sound and of high quality.

The key assertion is that the intensities of scattering from magnon annihilation and creation do not follow detailed balance. Certainly Fig 2(d) appears shows excess intensity for annihilation (negative energy) compared to the equilibrium state, whereas for creation (positive energy) the intensity looks very similar to the equilibrium state. The detailed analysis is done in Section II of the SI. Unfortunately, Fig S4 only has two of the four panels to which the caption and SI text refer. Fig. S4 shows only the laser excited data together with the models for equilibrium 3.6K and 4.6K and the residuals after subtracting those models from the excited data. This is what the caption to Fig. S4 refer to as panels (b) and (c) - and the text of the SI refer to as (b) and (d) (see lines 150 and 155) – and not (a) and (b) which is how the panels are labelled in the figure.

- Firstly, it is absolutely essential to show that equilibrium data are fully described by the MCViNE model at 3.6K, and the residuals are statistically consistent with zero. This is (i) to demonstrate that the MCViNE model (which includes resolution function effects? – please confirm) is accurate, and (ii) to enable the reader to judge the significance of the deviations from zero of the residuals of the laser-excited data that are shown in the current Fig. S4(b), judging by the legend. The data show over-subtraction at about +/- ~ 0.7 meV for annihilation and creation with respect to 4.6K (or 4.5K as the figure caption states). This is also what the SI text lines 158-161 states is in addition the case for the residuals of the equilibrium data with respect to the model for 4.6K. Without seeing both sets of data there is no way of knowing the significance. On the other hand,

is the legend in the current Fig. S4(b) incorrect and in fact it is showing the equilibrium data residuals and not the laser excited residuals as stated? The SI text lines 161-165 says that the residuals with respect to 4.6K are around zero for annihilation, which is not consistent with the figure. If this is the case, then the data are missing which supports the assertion in the text regarding the laser-excited data.

- Secondly, would a different effective temperature than 4.6K fit the data better – please explain this choice of temperature for the comparison, and/or perform a fit of $T_{\text{effective}}$ to the data? This is necessary regardless of which panels are actually present and missing in Fig. S4.

- Thirdly, please can the authors comment on the over-subtraction in the residuals comparison with both 3.6K and 4.6K for $E > 1$ meV?

- Fourthly, assuming Fig. S4(b) is for laser-excited data as the legend states, the present over-subtraction near 0.7 meV for creation seems to be almost as significant as the deviation for annihilation in terms of multiples of the error bar sizes. Is it possible that the spectrum satisfies (ignoring superscripts alpha and beta for clarity) $S(Q,-E) = \exp(-E/(k_B T)) S(Q,E)$ i.e. the data satisfy detailed balance, but that the population of magnons is not following the equilibrium Bose-Einstein distribution $n^{(BE)}(T)$? If that were the case, would it affect the interpretation of the data and theoretic discussion that follow? I strongly suspect it would.

Overall, there is a crucial uncertainty about the meaning of the data due to the missing panels, and the ambiguity of what data are actually being shown in the current Fig. S4(b). The reader (and this referee) need to see an unambiguous and full presentation of the data to be properly convinced that the data does indeed break detailed balance. The authors should address the other points above as part of re-editing. Without these data it is not possible to consider acceptance.

To support the assertion that the non-equilibrium state is steady, Fig S5 shows energy spectra for successive frames for excitation separation time $P = 51.7$ ms. The data seem to show clearly that there is no decrease of intensity in successive frames, However, it would be vastly preferable to present in addition the integrated intensity over the ranges used elsewhere in the paper e.g. Fig 3(a) and Fig. 3(b), to get a single data point with error bar for each frame, rather than the reader having to 'integrate by eye' - and construct an error bar by eye too. This should be done with the data for $P=61.8, 78.5, 95.2, P=262.2$ ms as well. Fig S6 is shows the existence of a non-equilibrium state (subject to revision convincing the reader about the primary matter raised above being addressed), but sums over all frames, and so does not demonstrate a steady state for each P .

There are a number of other editing/cross-referencing issues with the figures that need to be corrected.

- The integration range in H for Fig. 2(d) is stated as [0.49,0.51] in the caption. The data that are actually shown, 3.6K equilibrium data, are identical to Fig. S5, where it is stated as [0.475,0.525], and the data also look the same as Fig. S3, for which it is stated in the caption as [0.475, 0.525]. The integration range for Fig. S4a is not stated in the caption, but it must be narrower than for the equilibrium 3.6K data in Fig. S3, as the creation intensity spectrum is more sharply peaked. I suspect the caption to Fig. 2(d) is incorrect and the range is [0.49,0.51] for Fig. S4(a). Alternatively, the wrong panel has been placed at Fig. 2(d), which should be for the narrower integration range.

- SI line 119 refers to equilibrium data for 3.6K being shown in Fig 1(b); in fact it is Fig 2(a); the reference on line 120 to Figs S2(e)-(h) should be Figs S2(a)-(d).

- SI line 128 refers to Fig 1(d) but should be Fig 2(b)

- SI line 133, continuing the explanation of the data presented in that figure, says that the energy integration ranges are [-2,0.3] and [0.3, 2] meV, but the caption to Fig 2(b) states [-1.2, 0.3] and [0.3, 1.2].

- SI line 137: Again, Fig 1(d) should be Fig. 2(b)

- SI line 141 Fig 2(a) should be Fig 1(c).

- SI Fig S5 is the same as Fig 2(d). It is not clear why there needs to be repetition.

This may not be a complete list.

Some queries that need attention:

- SI Line 183: Quotes behaviour at $P = 251$ ms; should this be 262.2 ms? The rest of the paragraph and Fig S6 talks about $P=262.2$ ms.

- Please clarify how to interpret SI line 74 the statement "...an estimated temperature rise of 5K at the maximum laser incident energy...". Does this mean that the sample heats by 5K with each pulse, and if so, why is it that the annihilation and creation intensities are not correspondingly altered? Related to this, SI line 176-179 refers to a heating issue, meaning that

the number of pulses was reduced from 30 to 10 for the later experiment. This again raises the question of the effect of sample heating on the data analysis and the interpretation as a non-equilibrium steady state. These points need an explanation.

- SI line 137-138. The experimental data have a slope of unity, the model a little less. The sentence states that the deviation is due to the correction from the resolution function. If MCViNE is modelling the effect of the resolution function, then by definition shouldn't it be reproducing an effect that the experimental data be identically subject to as well?

- Main text line 150 says that magnon-phonon relaxation is inferred from the data to take place on the order of milliseconds, yet the SI line 271 says "...order of hundreds of milliseconds...". The two time scales have significance: the former is short on the time scale of the frame period, the latter long, and therefore on how a non-equilibrium steady state is achieved.

- Reference to Reeder et al PNAS 122 e2415300121 (2025) should be made.

Reviewer #3

(Remarks to the Author)

The authors reported a neutron scattering measurement on a classical 2D magnet Rb₂MnF₄ under laser irradiation, and discovered an interesting evolution of the magnon intensity on the anti-stokes side, which they argued to indicate a violation of the principle of detailed balance. The experiment is technically challenging, and the data quality is good (presumably due to the large Mn moment). More broadly, it opens up the possibility of studying nonequilibrium physics using neutron scattering, making the paper appropriate for Nature Communications. However, before I can recommend its publication, I would like to see the following addressed.

1. My biggest issue is with the interpretation of Fig 2b and Fig 3a. By plotting IS (Stokes intensity) vs IAS (Anti-Stokes intensity) and observing a slope not equal to 1, the authors argue that there is a violation of the principle of detailed balance in the non-equilibrium state.

I disagree with this interpretation, I think all Fig 2b and Fig 3a shows is that IS and IAS are no longer proportional to $n_{BE}(E,T)+1$ and $n_{BE}(E, T)$, respectively, when the system is out of equilibrium. However, the specific form of $I_S \propto n_{BE}+1$ and $I_{AS} \propto n_{BE}$ is only valid in a non-interacting magnon picture, but not a direct consequence of principle of detailed balance [$I_S/I_{AS}=\exp(E/kBT)$]. In other words, there are forms of IS and IAS satisfying detailed balance (a broader statement related to the reversibility of the system in equilibrium), but need not to strictly follow $n_{BE}+1$ and n_{BE} , respectively. For example, a simple (of course unphysical) $I_S=1$ and $I_{AS}=\exp(-E/kBT)$ satisfies detailed balance but will not give a slope of 1 when plotting IS vs IAS. Therefore, the only conclusion the authors can draw from Fig 3a is that a simple non-interacting picture no longer holds- but since the system is being pumped, this does not seem too surprising.

To definitively prove that detailed balance has been violated, one must therefore 1. independently measure the system temperature and 2. explicitly show $I_S/I_{AS} \neq \exp(E/kBT)$. A system (or lattice) temperature might be obtained from the IS and IAS of the phonon intensity – is this available in the authors data or is the experimentally measured Q too small to observe phonons?

2. Without explicitly measuring the system temperature in the non-equilibrium state (which might not be well-defined), an arguably better way to check for violation of detailed balance than that presented in the paper might be to plot the Q-resolved IS/IAS vs E(Q) at the SAME temperature and pump-probe delay [E(Q) is the experimentally measured magnon dispersion]. When the system is in equilibrium, IS/IAS can be described by $\exp(E(Q)/T)$ with a single T. On the other hand, this is no longer true when detailed balance is violated.

However, even disregarding potential difficulties associated with the poor signal-to-noise of the Q-dependent data, such a method still cannot definitively prove that detailed balance has been violated as magnon mode with different Q's might simply have different temperature when the system is out-of-equilibrium. This seems to be likely given that only the Q=0 magnon is pumped.

The following are more technical/minor

3. A well-known problem with using different pump and probe is the penetration depth mismatch. The authors partially addressed this by using a sample transparent to the pump wavelength. However, I noticed two potential caveats with their setup. First, the beam diameter is 5mm whereas the sample size is 10mm x 8mm- so only part of the probed sample volume is pumped. More importantly, the sample is illuminated not by a homogeneous laser spot, but a transient grating with a period of 2 μ m – why is a transient grating used? Does the obtained result depend on the grating period?

Consequently, the sample cannot be thought of as a homogeneously excited volume, but consists of (potentially coupled) excited and unexcited regions where the magnon intensities are likely very different; the sum of their responses is measured experimentally. Is inhomogeneity considered when arriving at the conclusion of 'violation of detailed balance'.

4. The data is a sum over all L. However, the relative orientations of pump and probe are different for different L. I am wondering whether the IS and IAS are L-independent? Could the authors verify this explicitly and include the L-resolved data in the Supplemental materials.

5. I completely do not understand the model given by Eq. (3). What is the bath mode, b? Is it lattice (phonon degrees of

freedom)? What justifies $\omega_s/\omega_b < 1$? If the 'bath' is supposed to mimic the thermal reservoir, which interacts with the sample and leads to thermal equilibration (represented by the coupling λ), why is a different damping, γ needed? What is the microscopic mechanism for γ ? Also, Eq. (3) does not seem to contain any periodic driving, which is crucial to the problem.

Overall, I expect a lot more discussions for the toy model, to better motivate it and make it more accessible to experimentalist like myself.

6. I would like the authors to highlight the clear distinction between a transient state ($P \gg \tau$ magnon-phonon- studied by a typical pump probe spectroscopy), and a steady state/periodic driving limit ($P \ll \tau$ magnon-phonon) studied by the present experiment. In particular, the authors should highlight the importance of the $1/P$ behaviour observed in Fig 3b- an indication of the absence of a well-defined time scale, and the absence of any decay in the AS signal even for the largest $P=262.2\text{ms}$ (Fig S7 should be moved to the main text). I almost missed this in my first reading- given the similarity of Fig 3b to an exponential decay with only 5 data points.

Overall, I think the manuscript presents interesting physics and significant technical advancement. Once the above (esp. 1 and 2) are addressed, I am happy to recommend its publication.

Version 1:

Reviewer comments:

Reviewer #2

(Remarks to the Author)

The authors have responded to all the comments in my review of the original submission, and I am now happy to recommend publication. The 'show-stopper' comments I had made were that the evidence for the breakdown of detailed balance had not been properly presented: (i) there were missing panels in the Supplementary Information (in Fig. S4, now Fig. S5) that meant that a reader could not assess in the data what the authors were arguing, and (ii) it was not clear to me that detailed balance breakdown and a change to magnon occupancy with energy could be resolved. The missing panels have now been added (and numerous other editing errors and references to the wrong figures have been corrected), and more careful elaboration in the text means that I now find their arguments convincing. In particular, the data Fig. S5 now clearly shows that the 3.6K equilibrium data is well described by the model for 3.6K (thereby setting the benchmark validation of the model) whereas the laser excited data is well described by the 3.6K equilibrium model on magnon creation but not on magnon annihilation (which is consistent with an effective 4.6K equilibrium model). This shows the breakdown of detailed balance. The authors have also addressed my concern that disentangling breakdown of detailed balance from possible magnon occupancy that no longer follows the Bose-Einstein form was not made, by adding a substantial discussion to the main article text. (This was added to address simultaneously the related concern of Referee 3.)

The lesser comments I made have all been addressed to my satisfaction, by small additions to the text, additional figures that I recommended, and correcting errors in the editing. Accordingly, I recommend acceptance.

Reviewer #3

(Remarks to the Author)

The authors' responses to my comments are excellent. I recommend the publication of the present work.

Revision Report for Manuscript Entitled:
 “Violation of detailed balance in non-equilibrium magnons observed by inelastic neutron scattering”
 (NCOMMS-25-77103-T)

We would like to thank the editor and referees for the review and constructive criticism. We considered and acted on all the recommendations, and below are our point-by-point responses to the detailed comments listed by the referees. All the changes in the main manuscript and Supplementary Information are highlighted as red text.

Referee #2

Overall comment: *The key assertion is that the intensities of scattering from magnon annihilation and creation do not follow detailed balance. Certainly Fig 2(d) appears shows excess intensity for annihilation (negative energy) compared to the equilibrium state, whereas for creation (positive energy) the intensity looks very similar to the equilibrium state. The detailed analysis is done in Section II of the SI. Unfortunately, Fig S4 only has two of the four panels to which the caption and SI text refer. Fig. S4 shows only the laser excited data together with the models for equilibrium 3.6K and 4.6K and the residuals after subtracting those models from the excited data. This is what the caption to Fig. S4 refer to as panels (b) and (c) - and the text of the SI refer to as (b) and (d) (see lines 150 and 155) – and not (a) and (b) which is how the panels are labelled in the figure.*

New Figure S5 caption: The measured intensities (solid red lines) (a) at an equilibrium temperature of 3.6 K and (b) with laser excitation are compared with the modeled intensity at both 3.6 K (dot-dashed black lines) and 4.6 K (dashed blue lines). L is integrated between [2.1, 2.9] (r.l.u) and H is integrated between [4.9, 5.1]. The residual intensity as a function of energy transfer E is calculated by subtracting the measured

intensity (c) at the equilibrium temperature and (d) with laser excitation with the modeled intensity at both 3.6 K (dot-dashed black lines) and 4.6K (dashed blue lines).

Response: we thank the referee for carefully reviewing our manuscript and for identifying the editing errors. We sincerely apologize for these mistakes. We have carefully re-examined the text and corrected all errors we were able to find.

The above plot is the updated Figure S5 in the current SI (formerly Fig. S4) with the caption in the red text.

Comment #1: *it is absolutely essential to show that equilibrium data are fully described by the MCViNE model at 3.6K, and the residuals are statistically consistent with zero. This is (i) to demonstrate that the MCViNE model (which includes resolution function effects? – please confirm) is accurate, and (ii) to enable the reader to judge the significance of the deviations from zero of the residuals of the laser-excited data that are shown in the current Fig. S4(b), judging by the legend. The data show over-subtraction at about +/- ~ 0.7 meV for annihilation and creation with respect to 4.6K (or 4.5K as the figure caption states). This is also what the SI text lines 158-161 states is in addition the case for the residuals of the equilibrium data with respect to the model for 4.6K. Without seeing both sets of data there is no way of knowing the significance. On the other hand, is the legend in the current Fig. S4(b) incorrect and in fact it is showing the equilibrium data residuals and not the laser excited residuals as stated? The SI text lines 161-165 says that the residuals with respect to 4.6K are around zero for annihilation, which is not consistent with the figure. If this is the case, then the data are missing which supports the assertion in the text regarding the laser-excited data.*

Response: We thank the referee for this careful and important observation. We fully agree that it is essential to demonstrate that the equilibrium data at all measured temperatures are accurately described by the MCViNE model and that the corresponding residuals are statistically consistent with zero. The modeled intensity already includes the effects of the instrumental resolution function, as stated in the Supplementary Information (lines 121–125). Indeed, a primary purpose of using MCViNE simulations is to accurately incorporate instrument resolution effects.

The updated Fig. S5(a) (formerly Fig. S4) presents a direct comparison between the measured intensity at 3.6 K and the modeled intensities at 3.6 K and 4.6 K. In this figure, L is integrated over [2.1, 2.9] r.l.u. and H over [4.9, 5.1]. The comparison shows that the equilibrium measurement at 3.6 K is accurately reproduced by the 3.6 K MCViNE model, whereas the 4.6 K model exhibits higher intensity on both the annihilation and creation sides. The residuals shown in Fig. S5(c) confirm this: the residuals obtained by subtracting the measured data from the 3.6 K model (dotted–dashed black line) fluctuate around zero within statistical uncertainty on both sides, while those obtained by subtracting the 4.6 K model from the 3.6 K data (dashed blue line) show nearly symmetric deviations on both sides.

To further demonstrate the accuracy of the model, we added an additional plot, Fig. S3 in the revised SI, also shown below. Beyond the colormaps in Fig. S2 and the integrated-Q cuts in Fig. S5, Fig. S3 compares cuts between the measured and modeled data at 3.6 K for a series of Q values. From Fig. S3(a) to (h), Q incrementally moves away from the magnetic zone center in steps of 0.005 r.l.u., which is the smallest step allowed by the instrumental resolution. This new figure clearly and unambiguously demonstrates that the MCViNE-simulated spectra well reproduce the measured equilibrium data.

New Figure S3 caption: Energy-dependent cuts between the measured (equilibrium; solid lines) and modeled (dashed lines) data at 3.6 K for a series of Q values. Q is along H in [HHL] and L is integrated between [2.1,2.9] in reciprocal lattice units (r.l.u.). From (a) to (h), Q incrementally moves away from the magnetic zone center in steps of 0.005 r.l.u.

The laser-excited measurements shown in Figs. S5(b) and S5(d) exhibit clear differences from the equilibrium data. As Fig. S5(b) demonstrates, while the measured creation intensity is well captured by the 3.6 K model, the measured annihilation intensity matches the 4.6 K model. This behavior is also reflected in the residual plots in Fig. S5(d). When the measured data are subtracted from the 3.6 K model (dotted-dashed black line), the residuals on the creation side fluctuate around zero within statistical uncertainty, whereas the annihilation side shows a pronounced excess residual. Conversely, when the measured data are subtracted from the 4.6 K model (dashed blue line), the annihilation-side residuals fluctuate around zero while the creation side exhibits excess intensity. The apparent over-subtraction near $E = -0.7$ meV, noted by the referee in Comment #4, lies within the statistical uncertainty and is highly asymmetric compared with the residuals on the creation side.

We have added the following sentences to the SI right below Eq. (4):

To demonstrate the accuracy of the model, Fig. S3 compares energy-dependent cuts between the measured and modeled data at 3.6 K for a series of Q values (Q is along H in [HHL]).

We have added the following sentences to the SI towards the end of page 9:

The apparent over-subtraction near $E = -0.6$ meV and under-subtraction near $E = -0.8$ meV and a slight over-subtraction over $E > 1$ meV (dashed blue line in Fig. S5(d)), may suggest that the shape of the magnon spectrum is modified under laser excitation. However, these features remain within statistical uncertainty, making it difficult to draw a definitive conclusion.

These corrections ensure that readers can clearly judge the significance of the deviations in the laser-excited residuals relative to the equilibrium baseline.

Comment #2: would a different effective temperature than 4.6K fit the data better – please explain this choice of temperature for the comparison, and/or perform a fit of $T_{\text{effective}}$ to the data? This is necessary regardless of which panels are actually present and missing in Fig. S4.

Response: The 4.6 K modeled data are chosen because they best reproduce the annihilation-side intensity of the laser-excited spectrum. The purpose of Fig. S5(b) is to illustrate that, under laser excitation, the creation and annihilation sides of the structure factor appear to correspond to two different effective temperatures. As we show in Fig. S5(d), the asymmetry associated with these two temperatures is statistically meaningful.

Whether the intensity under laser excitation should obey a Bose–Einstein distribution is a separate question. The apparent over-subtraction near $E = -0.6$ meV and under-subtraction near $E = -0.8$ meV (dashed blue line in Fig. S5(d)), as noted by the referee in Comment #4, may suggest that the shape of the annihilation spectrum is modified under laser excitation. However, these features remain within statistical uncertainty, making it difficult to draw a definitive conclusion.

We have added the following sentences to the SI between line 160-162:

While the modeled data at 3.6 K reproduce the creation-side intensity, the 4.6 K modeled data best reproduce the annihilation-side intensity of the laser-excited spectrum.

Comment #3: please can the authors comment on the over-subtraction in the residuals comparison with both 3.6K and 4.6K for $E > 1$ meV?

Response: The residuals above $E > 1$ meV show a small negative bias at both 3.6 K and 4.6 K that could be interpreted as “over-subtraction.” However, when the uncertainties are propagated (counting statistics and background subtraction), the deviation of the mean residual from zero in this energy range is not statistically significant: the residuals fluctuate around zero within their 1σ error bars and the apparent offset is comparable to the expected statistical scatter. We therefore do not find evidence for a systematic over-subtraction at $E > 1$ meV at either temperature.

Please refer to Comment #1 for details of the changes made in the Supporting Information.

Comment #4: assuming Fig. S4(b) is for laser-excited data as the legend states, the present over-subtraction near 0.7 meV for creation seems to be almost as significant as the deviation for annihilation in terms of multiples of the error bar sizes. Is it possible that the spectrum satisfies (ignoring superscripts alpha and beta for clarity) $S(Q, -E) = \exp(-E/(k_B T)) S(Q, E)$ i.e. the data satisfy detailed balance, but that the population of magnons is not following the equilibrium Bose-Einstein distribution $n^{(BE)}(T)$? if that were the case, would it affect the interpretation of the data and theoretic discussion that follow? I strongly suspect it would. Overall, there is a crucial uncertainty about the meaning of the data due to the missing panels, and the ambiguity of what data are actually being shown in the current Fig. S4(b). The reader (and this referee) need to see an unambiguous and full presentation of the data to be properly convinced that the data does indeed break detailed balance. The authors should address the other points above as part of re-editing. Without these data it is not possible to consider acceptance.

Response: This is precisely the motivation for Fig. S5(d) (formerly Fig. S4(b)). That panel demonstrates that a statistically meaningful over-subtraction is present on the creation side, whereas on the annihilation side the residuals obtained from the 4.6 K equilibrium data fluctuate around zero, consistent with statistical

noise. Furthermore, when compared with the equilibrium residuals shown in Fig. S5(c), the residuals in Fig. S5(d) exhibit highly asymmetric behaviors when subtracting the spectra modeled at 3.6 K and 4.6 K. This comparison highlights that the observed over-subtraction is not an artifact of the modeling procedure but reflects a genuine asymmetry between the creation and annihilation processes.

Regarding the reviewer's question of whether the spectrum can satisfy the detailed balance condition, $S(Q, -E) = \exp[-E/(k_B T)]S(Q, E)$, while not following an equilibrium Bose–Einstein distribution $n_{BE}(T)$, we emphasize that these two conditions are not independent. Detailed balance is a stringent requirement that applies only when a system is in thermodynamic equilibrium. For a magnon system, thermodynamic equilibrium necessarily implies a Bose–Einstein occupation, with a well-defined temperature, and thus satisfaction of detailed balance; the converse is also true. If the magnon population deviates from a Bose–Einstein distribution, the concept of a thermodynamic temperature becomes ill-defined and the system cannot be in equilibrium.

In our experiment, the magnon system is continuously driven by laser pumping and relaxes through dissipation into phonon baths, resulting in a nonequilibrium steady state rather than thermodynamic equilibrium. As discussed by M. J. Klein (Phys. Rev. 97, 6, 1955), nonequilibrium steady states sustained by cyclic driving processes cannot satisfy detailed balance. Consequently, we do not expect the spectrum to obey the detailed balance relation, $S(Q, -E) = \exp[-E/(k_B T)]S(Q, E)$ in the absence of a Bose–Einstein distribution, and we therefore conclude that a spectrum cannot satisfy detailed balance without simultaneously following an equilibrium Bose–Einstein occupation.

To emphasize this point, we have added an extensive discussion in the main text, beginning on page 6 (see also our response to Comment #1 of Referee 3).

Comment #5: *To support the assertion that the non-equilibrium state is steady, Fig S5 shows energy spectra for successive frames for excitation separation time $P = 51.7$ ms. The data seem to show clearly that there is no decrease of intensity in successive frames, However, it would be vastly preferable to present in addition the integrated intensity over the ranges used elsewhere in the paper e.g. Fig 3(a) and Fig. 3(b), to get a single data point with error bar for each frame, rather than the reader having to 'integrate by eye' - and construct an error bar by eye too. This should be done with the data for $P=61.8, 78.5, 95.2, P=262.2$ ms as well. Fig S6 is shows the existence of a non-equilibrium state (subject to revision convincing the reader about the primary matter raised above being addressed), but sums over all frames, and so does not demonstrate a steady state for each P .*

Response: We thank the reviewer for this comment. To further substantiate that the system reaches a steady state under periodic pumping, we have replaced the original Fig. S5 with a new figure (now Fig. S6). We note that the original Fig. S5 was identical to Fig. 2d in the main text; to avoid redundancy, it has therefore been replaced. Following the referee's suggestion, the new figure shows the integrated annihilation and creation intensity over four neutron frames. The laser excitation scheme corresponds to that shown in Fig. 2 of the main text.

New Figure S6 caption: Magnon annihilation (diamonds; left y axis) and creation (circles; right y axis) intensities measured over four neutron frames between two consecutive laser excitation events. No decay is observed across the four neutron frames, indicating steady-state behavior. The number of laser pulses per pumping event is set to $N=30$, and the number of neutron pulses incident on the sample between consecutive laser pumping events is $j = 4$, resulting in $P \sim 51.7$ ms. The data are integrated over $H \in [0.475, 0.525]$ and $L \in [2.1, 2.9]$ r.l.u.; Energy integration ranges are $E \in [-2, -0.3]$ meV for annihilation and $E \in [0.3, 2]$ meV for creation.

We have added the following sentences to the SI towards the end of page 10:

Figure S6 shows the integrated annihilation and creation intensity under laser excitation over four neutron frames between two consecutive laser excitation events. No decay is observed across the four neutron frames, indicating steady-state behavior.

To provide additional evidence for steady-state behavior, we have added the following plot to Fig. S8 (formerly Fig. S7). The original Fig. S7 contained only panel (a). The revised Fig. S8 not only compares the Q-integrated energy spectra at two different delay times [Fig. S8(a)], but also presents the integrated annihilation and creation intensities over sixteen neutron frames between two consecutive laser excitation events in Fig. S8(b).

New Figure S8 caption: (a) Comparison between the intensities with laser excitation at two delay times, $t_{\text{delay}} = 0$ ms (solid blue line) and 262.2 ms (red dashed line). Here, the delay time is defined as the difference in arrival time at the sample position between the last laser pulse in a laser excitation sequence and a particular neutron frame in that period. L is integrated between [2.1, 2.9] (r.l.u.) and H is integrated between [0.475, 0.525] (r.l.u.). (b) Magnon annihilation (diamonds; left y axis) and creation (circles; right y axis) intensities measured over sixteen neutron frames between two consecutive laser excitation events. The data are integrated over $H \in [0.475, 0.525]$ and $L \in [2.1, 2.9]$ (r.l.u.); Energy integration ranges are $E \in [-2, -0.3]$ meV for annihilation and $E \in [0.3, 2]$ meV for creation.

We have added the following sentences towards the end of Sec. IIB in the SI:

Figure S8(b) shows the integrated annihilation and creation intensities over sixteen neutron frames between two consecutive laser excitation events. No time-dependent behavior is observed within 262.2 ms.

Comment #6: *The integration range in H for Fig. 2(d) is stated as [0.49, 0.51] in the caption. The data that are actually shown, 3.6K equilibrium data, are identical to Fig. S5, where it is stated as [0.475, 0.525], and the data also look the same as Fig. S3, for which it is stated in the caption as [0.475, 0.525]. The integration range for Fig. S4a is not stated in the caption, but it must be narrower than for the equilibrium 3.6K data in Fig. S3, as the creation intensity spectrum is more sharply peaked. I suspect the caption to Fig. 2(d) is incorrect and the range is [0.49, 0.51] for Fig. S4(a). Alternatively, the wrong panel has been placed at Fig. 2(d), which should be for the narrower integration range.*

Response: we thank the referee for carefully reviewing our manuscript and for identifying the editing errors. We sincerely apologize for these mistakes. The H integration range in Fig. 2(d) is [0.475, 0.525]. We have now corrected this inconsistency in the text.

In Fig. S5 (formerly Fig. S4), H integration range in all the panels is [0.49, 0.51] and we have now clearly stated the integration range in the caption of Fig. S5. Fig. S5 is the only figure that uses a narrower H integration range. This choice was made to facilitate comparison with the modeled data, as the observation indicates that the excess intensity on the annihilation side under laser excitation is concentrated at the magnetic zone center.

Comment #7: *SI line 119 refers to equilibrium data for 3.6K being shown in Fig 1(b); in fact it is Fig 2(a); the reference on line 120 to Figs S2(e)-(h) should be Figs S2(a)-(d).*

SI line 128 refers to Fig 1(d) but should be Fig 2(b).

SI line 137: Again, Fig 1(d) should be Fig. 2(b)

SI line 141 Fig 2(a) should be Fig 1(c)

Response: We thank the referee for pointing out these editing errors and apologize for the oversight; they have now been corrected.

Comment #8: *SI line 133, continuing the explanation of the data presented in that figure, says that the energy integration ranges are [-2, 0.3] and [0.3, 2] meV, but the caption to Fig 2(b) states [-1.2, 0.3] and [0.3, 1.2].*

Response: We thank the referee for pointing out this inconsistency. The correct energy integration ranges are $[-2, -0.3]$ and $[0.3, 2]$, and this inconsistency has now been corrected. All the E-integrated data shown in the main text and SI have the same energy integration ranges.

Comment #9: *SI Fig S5 is the same as Fig 2(d). It is not clear why there needs to be repetition.*

Response: We thank the referee for pointing out this redundancy. As mentioned in the response to Comment #5, it has therefore been replaced. Following the referee's suggestion, the new figure shows the integrated annihilation and creation intensity over four neutron frames. The laser excitation scheme corresponds to that shown in Fig. 2 of the main text.

Comment #10: *SI Line 183: Quotes behaviour at $P = 251$ ms; should this be 262.2 ms? The rest of the paragraph and Fig S6 talks about $P=262.2$ ms.*

Response: We thank the referee for pointing out this error. P should be 262.2 ms instead of 251 ms. We have corrected this mistake in the SI.

Comment #11: *Please clarify how to interpret SI line 74 the statement "...an estimated temperature rise of 5K at the maximum laser incident energy...". Does this mean that the sample heats by 5K with each pulse, and if so, why is it that the annihilation and creation intensities are not correspondingly altered? Related to this, SI line 176-179 refers to a heating issue, meaning that the number of pulses was reduced from 30 to 10 for the later experiment. This again raises the question of the effect of sample heating on the data analysis and the interpretation as a non-equilibrium steady state. These points need an explanation.*

Response: The estimated 5 K temperature increase corresponds to a worst-case scenario in which the laser operates at maximum power and is continuously driven at 2000 Hz. For the laser-excited data presented in this work, both the laser power and excitation period were carefully tuned such that no excessive heating is observed in the INS data at the base temperature. Specifically, the intensity on the creation side remains unchanged compared with the 3.6 K equilibrium data, indicating negligible sample heating. Note that the laser power is adjusted by rotating a half-wave plate placed before a polarizing beam splitter.

The effects of sample heating would be clearly reflected in the magnon INS spectra. To illustrate this point, we have added a new figure to the Supplementary Information. The revised Fig. S7 compares laser-excited spectra obtained under low- and high-power laser excitation while keeping all other excitation parameters identical. Under low laser power, the sample remains at the 3.6 K base temperature, as evidenced by the unchanged creation-side intensity, which matches well with the 3.6 K equilibrium INS data, while an excess intensity is observed on the annihilation side. This behavior indicates the creation of a non-equilibrium magnon population induced by laser pumping.

In contrast, under high laser power, the intensities on both the creation and annihilation sides increase consistently with Bose–Einstein statistics and are well described by the 6 K equilibrium INS data. This behavior provides a clear signature of sample overheating.

New Figure S10 caption: Laser-excited magnon spectra (solid red lines) measured under (a) low- and (b) high-power laser excitation, with all other excitation parameters kept identical, compared to the equilibrium magnon spectra (dashed lines) measured at 3.6 K and 6 K. L is integrated between [2.1, 2.9] (r.l.u.) and H is integrated between [0.475, 0.525] (r.l.u.).

We have added a new section, Sec. IIC Overheating examples, to the SI with the following discussion:

An estimated temperature increase of 5 K, mentioned in Sec. IB, represents a worst-case operating condition, corresponding to continuous laser excitation at the maximum power and a repetition rate of 2000 Hz. In the laser-excited INS measurements discussed here, the laser power and excitation period were deliberately optimized to avoid such conditions. As a result, no evidence of excessive heating is observed at the base temperature. In particular, the intensity on the magnon creation side is unchanged relative to the 3.6 K equilibrium INS data, indicating negligible sample heating. The incident laser power was controlled by rotating a half-wave plate positioned upstream of a polarizing beam splitter.

Sample overheating is expected to manifest clearly in the magnon INS spectra. To demonstrate this effect, an additional figure is provided in the Supplementary Information (Fig. S10), which compares laser-excited spectra measured under low- and high-power laser excitation while keeping all other excitation parameters identical. Under low laser power, the sample remains at the base temperature of 3.6 K, as confirmed by the unchanged creation-side intensity that matches the 3.6 K equilibrium INS spectrum. At the same time, an excess intensity appears on the annihilation side, consistent with the formation of a non-equilibrium magnon population induced by laser pumping.

By contrast, under high laser power, the intensities on both the creation and annihilation sides increase and follow Bose–Einstein statistics. In this regime, the spectra are well described by the 6 K equilibrium INS data, providing a clear and unambiguous signature of sample overheating.

Comment #12: SI line 137-138. The experimental data have a slope of unity, the model a little less. The sentence states that the deviation is due to the correction from the resolution function. If MCViNE is modelling the effect of the resolution function, then by definition shouldn't it be reproducing an effect that the experimental data be identically subject to as well?

Response: We thank the referee for this point. It may not be appropriate to attribute the deviation from a unity slope solely to the resolution function, as the McViNE simulations already incorporate the effects of instrumental resolution. The experimental data at lower temperatures appear to follow a unity slope reasonably well. At higher temperatures, background noise may become more relevant, particularly for quantities involving integrated intensities, such as those shown in Fig. 2(b). To reflect this more cautious interpretation, we have revised the corresponding text in the manuscript to clarify that multiple factors, including background contributions, may influence the observed deviations.

We have added the following sentences in Sec. IIA of the SI:

The slight deviation of the slope from unity at elevated temperatures may originate from enhanced background noise levels that become more pronounced as the temperature increases.

Comment #13: *Main text line 150 says that magnon-phonon relaxation is inferred from the data to take place on the order of milliseconds, yet the SI line 271 says “...order of hundreds of milliseconds...”. The two time scales have significance: the former is short on the time scale of the frame period, the latter long, and therefore on how a non-equilibrium steady state is achieved.*

Response: We thank the referee for pointing out this inconsistency. Based on our non-equilibrium INS measurements and the Boltzmann transport analysis presented in Section SI.III, the magnon–phonon relaxation time is on the order of hundreds of milliseconds. The wording in the main text has been corrected accordingly.

Comment #14: *Reference to Reeder et al PNAS 122 e2415300121 (2025) should be made.*

Response: We thank the referee for pointing out this reference. It has now been cited toward the end of the main text.

Referee #3

Comment #1: *My biggest issue is with the interpretation of Fig 2b and Fig 3a. By plotting I_S (Stokes intensity) vs I_{AS} (Anti-Stokes intensity) and observing a slope not equal to 1, the authors argue that there is a violation of the principle of detailed balance in the non-equilibrium state.*

I disagree with this interpretation, I think all Fig 2b and Fig 3a shows is that I_S and I_{AS} are no longer proportional to $(n_{BE}(E,T)+1)$ and $n_{BE}(E, T)$, respectively, when the system is out of equilibrium. However, the specific form of $I_S \propto n_{BE}+1$ and $I_{AS} \propto n_{BE}$ is only valid in a non-interacting magnon picture, but not a direct consequence of principle of detailed balance [$I_S/I_{AS}=\exp(E/k_B T)$]. In other words, there are forms of I_S and I_{AS} satisfying detailed balance (a broader statement related to the reversibility of the system in equilibrium), but need not to strictly follow $n_{BE}+1$ and n_{BE} , respectively. For example, a simple (of course unphysical) $I_S=1$ and $I_{AS}=\exp(-E/k_B T)$ satisfies detailed balance but will not give a slope of 1 when plotting I_S vs I_{AS} . Therefore, the only conclusion the authors can draw from Fig 3a is that a simple non-interacting picture no longer holds- but since the system is being pumped, this does not seem too surprising.

To definitively prove that detailed balance has been violated, one must therefore 1. independently measure the system temperature and 2. explicitly show $I_S/I_{AS} \neq \exp(E/k_B T)$. A system (or lattice) temperature might be obtained from the I_S and I_{AS} of the phonon intensity – is this available in the authors data or is the experimentally measured Q too small to observe phonons?

Without explicitly measuring the system temperature in the non-equilibrium state (which might not be well-defined), an arguably better way to check for violation of detailed balance than that presented in the paper might be to plot the Q -resolved I_S/I_{AS} vs $E(Q)$ at the SAME temperature and pump-probe delay [$E(Q)$ is the experimentally measured magnon dispersion]. When the system is in equilibrium, I_S/I_{AS} can be described by $\exp(E(Q)/T)$ with a single T . On the other hand, this is no longer true when detailed balance is violated.

However, even disregarding potential difficulties associated with the poor signal-to-noise of the Q -dependent data, such a method still cannot definitively prove that detailed balance has been violated as magnon mode with different Q 's might simply have different temperature when the system is out-of-equilibrium. This seems to be likely given that only the $Q=0$ magnon is pumped.

Response: We thank the referee for the insightful conceptual distinction between the detailed balance ratio ($I_S/I_{AS}=\exp(E/k_B T)$) and the non-interacting magnon distribution, n_{BE} and $n_{BE}+1$. A similar concern was also raised by another referee, namely whether the spectrum can satisfy the detailed balance condition, $S(Q,-E) = \exp[-E/(k_B T)]S(Q, E)$ without following an equilibrium Bose–Einstein distribution $n_{BE}(T)$. While we agree that, in principle, these two conditions can be mathematically decoupled, we emphasize that for the bosonic system studied here they are fundamentally linked.

Detailed balance is a stringent condition that applies only when a system is in thermodynamic equilibrium. For a bosonic magnon system, thermodynamic equilibrium necessarily entails a Bose–Einstein occupation characterized by a single, well-defined temperature, which in turn enforces detailed balance between magnon creation and annihilation processes. Crucially, the converse is also true: if the magnon population deviates from a Bose–Einstein distribution, the concept of a thermodynamic temperature becomes ill-defined, indicating that the system is not in equilibrium and that detailed balance cannot hold.

We therefore disagree with the referee’s assertion that a definitive demonstration of broken detailed balance must rely exclusively on (i) an independent measurement of the system temperature and (ii) an explicit violation of $I_S/I_{AS} = \exp(E/k_B T)$. This criterion is incomplete for an out-of-equilibrium bosonic system, where a unique, well-defined temperature may not exist. In such cases, the apparent satisfaction of the detailed-balance ratio can be incidental and physically meaningless. Demonstrating that different parts of the spectrum imply incompatible effective temperatures, or that the temperature itself becomes ill-defined, constitutes equally strong—and in fact more general—evidence for the breakdown of detailed balance.

From the outset of our systematic analysis, we carefully considered all the alternative scenarios, as also suggested by the referee, and explored multiple approaches to identify unambiguous evidence of broken detailed balance. After thorough examination, we concluded that the present method of data presentation provides the most compelling and direct evidence for the breakdown of detailed balance between magnon creation and annihilation processes. Below, we provide a detailed justification for our approach.

- In principle, one may test whether detailed balance is violated by examining whether the relation $I_S/I_{AS} = \exp(E/k_B T)$ holds. In practice, however, this approach becomes unreliable at low temperatures. When the dispersion is sharp, most of the spectral weight is concentrated at the lowest energies, and the measured lineshape is largely dominated by instrumental energy resolution rather than intrinsic magnon properties. As a result, only the peak intensity carries meaningful information, while the overall spectral shape provides little additional constraint. Consequently, the fitting procedure is effectively governed by matching the exponential factor $\exp(E/k_B T)$ to the peak (or near-peak) intensity alone. Because this constitutes a single-point constraint, it is always possible to extract an “effective temperature” that satisfies the relation. However, such a fitted temperature has limited physical significance once the magnon system deviates from an equilibrium Bose–Einstein distribution.

Figure 1: Energy-dependent cuts measured (a) at equilibrium and (b) under laser excitation, respectively. The detailed-balance fitting procedure was carried out by reflecting the spectrum $I(E)$ to obtain $I(-E)$ and then multiplying $I(-E)$ by $\exp(E/k_B T)$, with T treated as a fitting parameter. The resulting spectrum (dashed lines) was fitted to the original $I(E)$. (c) Direct comparison between the energy-dependent cuts measured at 3.6 K (dashed line) and under laser excitation (solid line).

To illustrate this point, we present an example from our experimental data. Panels (a) and (b) show the energy-dependent intensities (solid lines) measured at equilibrium and under laser excitation, respectively. The detailed-balance fitting procedure was carried out by reflecting the spectrum $I(E)$ to obtain $I(-E)$, and then multiplying $I(-E)$ by $\exp(E/k_B T)$, with T treated as a fitting parameter. The resulting spectrum (dashed lines) was fitted to the original $I(E)$.

For the equilibrium measurement, the fitted temperature is 3.6K, consistent with the sample temperature. For the laser-excited measurement, a similarly good fit can still be obtained, yielding an effective temperature of approximately 4.6K. At first glance, this suggests that the detailed-balance relation $I_S/I_{AS} = \exp(E/k_B T)$ remains satisfied even under laser excitation.

However, a direct comparison of the energy-dependent spectra reveals a clear deviation [panel (c)]. The spectra on the magnon-creation side are essentially identical for the equilibrium and laser-excited measurements, whereas on the annihilation side the laser-excited spectrum exhibits a systematically higher intensity than the equilibrium spectrum. If one were to naively interpret the laser-excited data using a Bose–Einstein distribution, this would imply that the creation process remains characterized by a temperature of 3.6 K, while the annihilation process corresponds to a higher effective temperature. This discrepancy suggests that under interacting conditions, the magnons deviate from equilibrium statistics and the system does not have a well-defined temperature, rendering both the 'effective temperature' and the detailed balance equality itself physically meaningless.

The referee provides a hypothetical example ($I_S = 1$ and $I_{AS} = \exp(E/k_B T)$) that satisfies detailed balance but implies a non-bosonic distribution. We emphasize that our system remains fundamentally bosonic. Our measurements demonstrate that the magnetic Bragg peaks, magnon dispersion, and Stokes intensity (I_S) remain unchanged under laser excitation. Since I_S continues to reflect standard magnon creation processes at equilibrium, the observed deviation in the I_S / I_{AS} ratio must stem from an excess population on the anti-Stokes side that no longer follows a Bose-Einstein distribution. Because transition rates are proportional to these occupation numbers, this population asymmetry implies that creation and annihilation processes are no longer microscopically reversible. Consequently, one can conclude that the detailed balance is broken.

In our experiment, the magnon system is continuously driven by laser pumping and relaxes through dissipation into phonon baths, resulting in a nonequilibrium steady state rather than thermodynamic equilibrium. As discussed by M. J. Klein (Phys. Rev. 97, 6, 1955), nonequilibrium steady states

sustained by cyclic driving processes cannot satisfy detailed balance. Consequently, we do not expect the spectrum to obey the detailed balance relation, $S(Q, -E) = \exp[-E/(k_B T)]S(Q, E)$ in the absence of a Bose–Einstein distribution, and we therefore conclude that a spectrum cannot satisfy detailed balance without simultaneously following an equilibrium Bose–Einstein occupation for a bosonic system.

- As part of our systematic analysis, we examined the energy-dependent spectra at equilibrium and under laser excitation across multiple momentum-transfer Q points (shown in Fig. 2 on the next page). This analysis reveals that the nonequilibrium population is strongly concentrated near the Brillouin-zone center. In particular, the excess intensity on the magnon-annihilation side is most pronounced at and around $Q = 0.5$ r.l.u [Fig. 2(a)-(c)]. As Q moves away from the zone center, the excess intensity progressively diminishes, and at sufficiently large Q the laser-excited and equilibrium spectra become indistinguishable on the annihilation side [Fig. 2(d)-(e)].

We note that away from the zone center the overall annihilation-side intensity decreases substantially, leading to a reduced signal-to-noise ratio. Consequently, any apparent differences between equilibrium and laser-excited spectra become increasingly difficult to resolve as illustrated in Fig. 3 below, and extracting an effective temperature from the ratio I_S/I_{AS} at those large Q points becomes unreliable.

Figure 2: Energy-dependent cuts measured at 3.6 K and under laser excitation. The laser excitation scheme is the same as presented in Fig. 1 above. From (a) to (f), Q incrementally moves away from the magnetic zone center in steps of 0.005 r.l.u.

Figure 3: Total annihilation intensity as a function of H in [HHL] (r.l.u) under three different laser excitation schemes compared to that measured at 3.6 K.

- We agree that showing the true lattice temperature using phonon dispersion would be ideal. Unfortunately, we could not obtain the phonon dispersion due to Q-range constraints resulting from mounting the laser setup onto the beamline. Under these constraints, we performed extensive equilibrium measurements on the magnon dispersion at various temperatures. The observed deviation from the intensity and shape of the dispersion under pumping conditions serves as our primary indicator that the magnon system is out-of-equilibrium.

We realized that the previous presentation involved a substantial conceptual leap from nonequilibrium statistics to the broken detailed balance condition $S(Q, -E) \neq \exp[-E/(k_B T)]S(Q, E)$. To bridge this gap, we have added an extensive discussion in the main text, beginning on page 6, as follows:

To investigate this effect in greater detail, we integrate the signal over a narrow H -range around the zone center. Fig. 2(d), which shows data separately reduced for each neutron frame between the laser pumping events, reveals that the magnon creation side is well described by the equilibrium data, while a photoinduced excess intensity appears only on the annihilation side. As shown in SI Sec. IIB, this increase in annihilation intensity is statistically significant and non-thermal in origin.

This energy-resolved analysis reveals a clear breakdown of equilibrium magnon statistics. A natural question is whether the condition of detailed balance could nevertheless remain valid when the magnon occupation deviates from a Bose–Einstein distribution. Although these two conditions can, in principle, be mathematically decoupled, we emphasize that for the bosonic magnon system studied here they are fundamentally linked.

Assessing detailed balance solely through the relation, $I(Q, -E) \neq \exp(-E/k_B T)I(Q, E)$ is insufficient in an out-of-equilibrium bosonic system. When a unique thermodynamic temperature does not exist, apparent agreement with this relation can arise trivially—particularly at low temperatures, where spectral weight is concentrated near the dispersion minimum and the measured lineshape is dominated by instrumental energy resolution rather than intrinsic magnon statistics. In this regime, fitting the intensity ratio effectively reduces to a single-point constraint, making it always possible to extract an “effective temperature” irrespective of whether the underlying magnon population is thermal. Such a temperature lacks physical meaning once the system departs from equilibrium.

In equilibrium, a bosonic magnon system is described by a Bose–Einstein distribution characterized by a single, well-defined temperature, which enforces detailed balance between magnon creation

and annihilation processes. Under laser excitation, however, the magnon-creation spectra remain unchanged, while the annihilation intensity is systematically enhanced. Interpreted within an equilibrium framework, this would require mutually incompatible temperatures for creation and annihilation processes, directly demonstrating that the magnon population cannot be described by a single Bose–Einstein distribution.

Moreover, as shown in SI Sec. IIB, not only the magnon-creation spectra but also the magnetic Bragg peaks remain unchanged under laser excitation, confirming that the long-range magnetic order is preserved and that the system remains fundamentally bosonic. The excess intensity on the annihilation side therefore reflects a nonthermal overpopulation of magnons. Since transition rates are proportional to occupation numbers, this population asymmetry implies that magnon creation and annihilation processes are no longer microscopically reversible. We thus conclude that the definition of a thermodynamic temperature becomes ill-defined under laser excitation and, consequently, that the dynamic magnetic structure factor violates the condition of detailed balance, i.e. $I(Q, -E) \neq \exp(-E/k_B T) I(Q, E)$.

Comment #2: *A well-known problem with using different pump and probe is the penetration depth mismatch. The authors partially addressed this by using a sample transparent to the pump wavelength. However, I noticed two potential caveats with their setup. First, the beam diameter is 5mm whereas the sample size is 10mm x 8mm- so only part of the probed sample volume is pumped. More importantly, the sample is illuminated not by a homogeneous laser spot, but a transient grating with a period of $2\mu\text{m}$ – why is a transient grating used? Does the obtained result depend on the grating period?*

Consequently, the sample cannot be thought of as a homogeneously excited volume but consists of (potentially coupled) excited and unexcited regions where the magnon intensities are likely very different; the sum of their responses is measured experimentally. Is inhomogeneity considered when arriving at the conclusion of ‘violation of detailed balance’.

Response: We thank the referee for this thoughtful comment. Indeed, the sample was illuminated using a grating pattern with a spot size of approximately 5 mm in diameter. When we initially designed the experiment, we intended to use this grating-pattern illumination—with a grating period comparable to the magnon mean free path—to induce nonequilibrium magnon transport, analogous to previous transient grating experiments on phonon transport. However, one aspect we did not fully consider at the time was the extremely weak coupling between magnons and phonons in this class of materials, which leads to a very long magnon–phonon coupling time.

Turning to the referee’s question of whether inhomogeneity arising from nonuniform illumination could affect our conclusion regarding the violation of detailed balance, our answer is no. This issue is also closely related to the referee’s later comments concerning modeling and the distinction between transient and steady-state driving.

The observed violation of detailed balance is robust against initial spatial inhomogeneities in the optical excitation due to a clear separation of relaxation timescales. While electron and phonon subsystems reach equilibrium on sub-microsecond to microsecond scales—confirmed by an undistorted magnon-creation spectrum and thermal modeling—the magnon subsystem exhibits a significantly slower relaxation. In gapped Heisenberg antiferromagnets such as Rb_2MnF_4 , dominant magnon–magnon scattering is restricted to number-conserving two-in/two-out processes that redistribute population toward the Brillouin zone

center without restoring a thermal distribution. The spectral gap creates a relaxation bottleneck, where further equilibration depends on the significantly longer timescales of magnon–phonon coupling. Consequently, when the driving period is shorter than the magnon–phonon coupling time, the system enters a non-equilibrium steady state governed by conservation laws rather than lattice temperature. The detailed balance violation reflects this driven steady state, independent of the initial excitation profile.

We have also added the following sentence toward the end of SI Sec. IB:

Note that initial spatial inhomogeneities do not affect the observed violation of detailed balance, as the violation is an intrinsic feature of the driven steady state rather than a transient effect of the initial excitation as discussed in the main text.

Comment #3: The data is a sum over all L . However, the relative orientations of pump and probe are different for different L . I am wondering whether the I_S and I_{AS} are L -independent? Could the authors verify this explicitly and include the L -resolved data in the Supplemental materials.

Response: We thank the referee for the suggestion. As Rb_2MnF_4 is a quasi-2D Heisenberg antiferromagnet, magnon transport is confined to the ab -plane, with negligible coupling. Figure below shows the INS intensity maps along L direction measured at 3.6 K and under laser excitation and confirms no observable dispersion along the L direction with and without laser excitation.

Figure 4: INS intensity maps along L direction measured (a) at 3.6 K and (b) under laser excitation. H is integrated between 0.475 and 0.525.

Comment #4: I completely do not understand the model given by Eq. (3). What is the bath mode, b ? Is it lattice (phonon degrees of freedom)? What justifies $\omega_s/\omega_b < 1$? If the ‘bath’ is supposed to mimic the thermal reservoir, which interacts with the sample and leads to thermal equilibration (represented by the coupling λ), why is a different damping, γ needed? What is the microscopic mechanism for γ ? Also, Eq. (3) does not seem to contain any periodic driving, which is crucial to the problem. Overall, I expect a lot more discussions for the toy model, to better motivate it and make it more accessible to experimentalist like myself.

Response: we thank the referee for this comment. In the model, the bath mode, b , represents the lattice degree of freedom, while δa corresponds to the spin degree of freedom. The assumption $\omega_s/\omega_b < 1$ is introduced purely for analytical convenience, indicating that the energy of the bath mode exceeds that of

the spin mode. This hierarchy allows for a controlled expansion of the transformation matrix S , enabling us to neglect higher-order terms in $\tilde{\omega}_s/\omega_s$ and to obtain closed-form analytical expressions for the eigenmodes. Importantly, this assumption is not intended to impose a physical constraint. In realistic systems, multiple bath and spin modes are present, and no simple one-to-one energy relation applies. Diagonalization of the coupled Hamiltonian yields the eigenmodes of the interacting system. Independent of the specific assumption, $\omega_s/\omega_b < 1$, the normal modes, c and c^\dagger , are expressed as a linear combination of b , b^\dagger , δa , and δa^\dagger . While the assumption modifies the numerical values of the coefficients in these linear combinations, it does not alter the formal structure of the resulting equations.

This minimal toy model is designed to illustrate the evolution of creation and annihilation rates in a coupled system subject to external driving. The parameter λ denotes the coupling strength between the spin and lattice modes; this coupling hybridizes the original degrees of freedom and gives rise to new collective quasiparticles that characterize the long-range interacting system. Only after identifying these eigenmodes do we introduce generation of the quasiparticles and their damping at rate γ , which governs relaxation toward thermal equilibrium. The parameters λ and γ therefore describe distinct physical processes and have different microscopic origins.

To better motivate our toy model, we have now modified the paragraph after Eq. (3) as following:

Equation 3 describes two coupled single-mode harmonic oscillators: a "spin" mode, $\hbar\omega_s\delta a^\dagger\delta a$, and a "bath" mode, $\hbar\omega_b b^\dagger b$, representing spin and lattice degree of freedom, respectively. The parameter λ denotes the coupling strength between the two modes; this coupling hybridizes the original degrees of freedom and gives rise to new collective quasiparticles that characterize the long-range interacting system. Only after identifying these eigenmodes do we introduce external pumping of the quasiparticles and their damping at rate γ , which governs slow relaxation toward thermal equilibrium. Although highly simplified, this minimal toy model captures the essential physics of driven dissipative dynamics at a heuristic level.

We have also added the following discussion about the assumption $\omega_s/\omega_b < 1$ in the Sec. IV of the SI:

The assumption $\omega_s/\omega_b < 1$ is introduced purely for analytical convenience, indicating that the energy of the bath mode exceeds that of the spin mode. This hierarchy allows for a controlled expansion of the transformation matrix S , enabling us to neglect higher-order terms in $\tilde{\omega}_s/\omega_s$ and to obtain closed-form analytical expressions for the eigenmodes. Importantly, this assumption is not intended to impose a physical constraint. In realistic systems, multiple bath and spin modes are present, and no simple one-to-one energy relation applies. Diagonalization of the coupled Hamiltonian yields the eigenmodes of the interacting system. Independent of the specific assumption, $\omega_s/\omega_b < 1$, the normal modes, c and c^\dagger , are expressed as a linear combination of b , b^\dagger , δa , and δa^\dagger . While the assumption modifies the numerical values of the coefficients in these linear combinations, it does not alter the formal structure of the resulting equations.

Comment #5: *I would like the authors to highlight the clear distinction between a transient state ($P > \tau_{\text{magnon-phonon}}$ studied by a typical pump probe spectroscopy), and a steady state/periodic driving limit ($P \ll \tau_{\text{magnon-phonon}}$) studied by the present experiment. In particular, the authors should highlight the importance of the $1/P$ behavior observed in Fig 3b- an indication of the absence of a well-defined time scale, and the absence of any decay in the AS signal even for the largest $P=262.2\text{ms}$ (Fig S7 should be*

moved to the main text). I almost missed this in my first reading- given the similarity of Fig 3b to an exponential decay with only 5 data points.

Response: We thank the referee for this comment. We agree that the discussion of the 1/P behavior is essential for understanding the experimental results. Although the nonequilibrium steady-state interpretation and the associated 1/P scaling were already discussed in the main text (see pages 9–10), we have further improved the presentation in the revised manuscript. Specifically, we have simplified Fig. 3 to panels (a) and (b) only, and we now explicitly emphasize in the figure caption that the observed 1/P behavior is a signature of a nonequilibrium steady state in a driven–dissipative system. In addition, we have added a new figure (Fig. 4), together with a corresponding discussion toward the end of page 11:

The separation of relaxation timescales—illustrated in Fig. 4(a)—facilitates the observed non-equilibrium steady state. While electron and phonon subsystems thermalize on submicrosecond to microsecond scales (verified by an undistorted magnon-creation spectrum and thermal modeling), the magnon subsystem remains decoupled due to significantly slower relaxation. In gapped Heisenberg antiferromagnets like Rb_2MnF_4 , number-conserving two-in/two-out magnon scattering redistributes population toward the Brillouin zone center but cannot restore a thermal distribution. The spectral gap acts as a relaxation bottleneck, deferring equilibration to the much longer timescales of magnon–phonon coupling. Because the driving period is shorter than this coupling time, the system enters a steady state governed by conservation laws rather than lattice temperature. Consequently, the violation of detailed balance is an intrinsic feature of the driven magnon population, independent of initial excitation inhomogeneities.

Main text new Figure 4 caption: (a) Timeline of physical events in a laser-neutron pump-probe experiment. The intense pulsed laser induces hot electrons and spins of the excited electrons are then affected, leading to non-equilibrium spin states within nanoseconds. Through elastic pairwise collisions of magnons, within a few microseconds at low temperatures, high energy non-equilibrium magnons decay to the lowest energy states with a

non-equilibrium magnon distribution in the form of $\left[\exp\left(\frac{E-\mu^{ss}}{k_B T_{ex}}\right) - 1 \right]^{-1}$, where μ^{ss} is a chemical potential. (b)

Illustration of out-of-equilibrium processes during INS. The population distribution of the excited states is a result of laser excitation, while the ground state remains intact and still in equilibrium with the thermal bath. When neutrons annihilate magnons, they detect the population distribution of non-equilibrium magnon states induced by the laser. When neutrons create magnons from the ground state, they create a canonical ensemble average of magnons according to the temperature of the ground state.